# EXPLORING THE FORGETTING IN ADVERSARIAL TRAINING: A NOVEL METHOD FOR ENHANCING ROBUSTNESS

**Xianglu Wang**[1]**, Hu Ding**[2*]
[1]School of Artificial Intelligence and Data Science, [2]School of Computer Science and Technology
University of Science and Technology of China
{wxlu}@mail.ustc.edu.cn, {huding}@ustc.edu.cn

## ABSTRACT

In recent years, there has been an explosion of research into developing robust deep neural networks against adversarial examples. As one of the most successful methods, *Adversarial Training (AT)* has been widely studied before, but there is still a gap to achieve promising clean and robust accuracy for many practical tasks. In this paper, we consider the AT problem from a new perspective which connects it to *catastrophic forgetting* in continual learning (CL). Catastrophic forgetting is a phenomenon in which neural networks forget old knowledge upon learning a new task. Although AT and CL are two different problems, we show that they actually share several key properties in their training processes. Specifically, we conduct an empirical study and find that this forgetting phenomenon indeed occurs in adversarial robust training across multiple datasets (SVHN, CIFAR-10, CIFAR-100, and TinyImageNet) and perturbation models ($\ell_\infty$ and $\ell_2$). Based on this observation, we propose a novel method called **A**daptive **M**ulti-teachers **S**elf-distillation (**AMS**), which leverages a carefully designed adaptive regularizer to mitigate the forgetting by aligning model outputs between new and old "stages". Moreover, our approach can be used as a unified method to enhance multiple different AT algorithms. Our experiments demonstrate that our method can significantly enhance robust accuracy and meanwhile preserve high clean accuracy, under several popular adversarial attacks (e.g., PGD, CW, and Auto Attacks). As another benefit of our method, we discover that it can largely alleviate the robust overfitting issue of AT in our experiments.

## 1 INTRODUCTION

Deep Neural Networks (DNNs) (LeCun et al., 2015) have demonstrated state-of-the-art performance in a number of fields, such as computer vision (He et al., 2016) and natural language processing (Devlin et al., 2019). But DNNs have been shown to be vulnerable to adversarial perturbation examples. Such examples are crafted by making small changes to natural data, but these changes are often imperceptible to human observers (Goodfellow et al., 2015; Szegedy et al., 2014). This vulnerability of DNNs raises significant security concerns regarding their practicality in security-sensitive applications, such as face recognition (Parkhi et al., 2015) and autonomous driving (Chen et al., 2015).

To address the security concerns, a great deal of defense methods have been developed to improve the adversarial robustness of DNNs (Buckman et al., 2018; Guo et al., 2018; Song et al., 2018; Xie et al., 2018; Madry et al., 2018; Zhang et al., 2019b; Yang et al., 2024; Dedeoglu et al., 2024). Among existing defense strategies, **Adversarial Training (AT)** (Madry et al., 2018; Zhang et al., 2019b) has been demonstrated as one of the most effective methods to defend against adversarial examples (Pang et al., 2021; Maini et al., 2020). The idea of AT is utilizing adversarial samples to train the model in each training epoch, which can be formulated as a "minimax robust optimization" problem, searching for the best solution to the worst-case scenario. However, the robust accuracies achieved by AT are still not that satisfying for many tasks. For instance, as a representative AT method, if we perform TRADES (Zhang et al., 2019b) with the $\ell_\infty$ norm on CIFAR-10, it yields a

---

*Corresponding author.

robust accuracy less than 50% under Auto Attack. A complete introduction of existing AT methods is provided in Section 2.

In this paper, we conduct a deep investigation on the learning process of adversarial samples, because whether the model can effectively learn adversarial examples greatly determines the final adversarial training performance. In particular, we build the seemingly unusual connection between AT and Continual learning (CL), an area that has been extensively studied recently (Wang et al., 2024). Our inspiration comes from the following observation: during the adversarial training process, the distribution of adversarial examples varies across different training epochs, a phenomenon quite similar to learning different distributions in CL (each distribution corresponds to a unique learning task). Specifically, the standard DNNs training process implicitly assumes that the data are drawn independently and identically distributed (i.i.d.) from the same probability distribution. However, in the scenario of AT, the generated adversarial samples often do **not** hold this assumption. Note that the adversarial samples in each epoch are usually generated by gradient information determined by current model parameter (Madry et al., 2018). Therefore, as the model parameter updates during the training process, the distribution of adversarial samples should continuously shift over time. Thus, each stage of the alternating training procedure can be regarded as a unique task (similar with a CL scenario). In addition, unlike conventional machine learning models built on static data distribution, CL is characterized by learning from dynamic data distributions and often suffers from *catastrophic forgetting* (Kirkpatrick et al., 2017). Specifically, the catastrophic forgetting is a phenomenon in which neural networks forget old knowledge upon learning a new task. Considering the similarity between the AT and CL problems, it is natural to ask the following questions:

*Does the forgetting phenomenon also occur in adversarial training process? If so, is it possible to improve the performance of adversarial training through mitigating the forgetting?*

## 1.1 OUR MAIN CONTRIBUTIONS

Firstly, we conduct an empirical study to answer the first question. Our experimental results uncover that the forgetting phenomenon indeed appears in adversarial training process. Specifically, when the model is trained in the later epochs, it is possible to forget the information from adversarial samples learned in earlier epochs. For instance, as shown in Figure 1, we continuously monitor the classification probability changes of an adversarial sample with the true label "ship" during the adversarial training process (the total epochs are evenly divided into 5 stages). We find that during stages 1 to 3, the sample was correctly classified and the confidence score reached 41. 79% in stage 2. However, the sample was misclassified in stages 4 and 5, and the confidence had dropped to merely 7.15% in stage 5. In our empirical study, we can also observe this phenomenon on multiple datasets beyond CIFAR-10, such as SVHN, CIFAR-100, and TinyImageNet. It is worth noting that the previous work (Gupta et al., 2020) illustrated another experiment, using the adversarial samples generated at different training stages to test the final built model. As we know, an adversarial sample is generated based on the current parameters and gradient information of the training stage it belongs to, so their experimental purpose is not about uncovering the "forgetting" issue of a fixed adversarial example across different training stages. On the other hands, our used adversarial sample in Figure 1 is generated at the beginning, and then is always fixed to test across all the stages (so as to illustrate the forgetting towards this fixed adversarial sample).

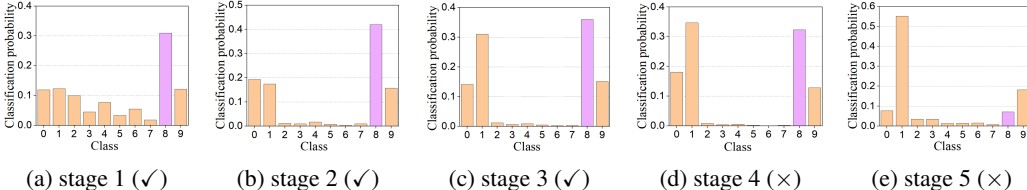

| (a) stage 1 (✓) | (b) stage 2 (✓) | (c) stage 3 (✓) | (d) stage 4 (×) | (e) stage 5 (×) |

Figure 1: Demonstration of a real adversarial example being learned and correctly classified by the model in Stage 1, but forgotten and misclassified in stage 5. The true label of this adversarial example is "ship". The AT method is TRADES (Zhang et al., 2019b). The horizontal axis represents the ten classes in CIFAR-10 and the vertical axis represents classification probabilities. Probabilities corresponding to the true label are shown in purple, while all others are shown in orange. "✓" means "classified correctly", and "×" means "classified wrongly".

Following these empirical findings, we try to explore that whether the performance of AT can be enhanced by mitigating the forgetting issue. In the line of research on CL, there have been numerous methods emerged for addressing the phenomenon of catastrophic forgetting, such as the replay-based methods (Buzzega et al., 2020; Boschini et al., 2022) and the regularization-based methods (Sun et al., 2023; Rebuffi et al., 2017). However, most of these methods are designed for standard CL tasks and cannot be directly applied to AT, due to the following two major challenges. **(1)** For AT, the model should capture much more subtle information from each adversarial example, compared to ordinary image classification problems, because an adversarial sample is usually generated by adding small and carefully crafted changes. For example, perturbations may target specific features such as a cat's tail, rather than altering the entire sample's classification from cat to dog. **(2)** As we known, adversarial training is inherently much harder than training a standard DNN, as theoretically proved in (Zhang et al., 2019b). For example, adversarial training models usually have a low classification accuracy (e.g., roughly less than 40% on CIFAR-100), which may lead to transferring a certain extent of incorrect information to the subsequent "stages". Thus, how to transfer the previous correct information and avoid the negative impact from the incorrect information, is another obstacle that hinders directly applying existing CL methods to enhance AT.

To address these challenges, we propose a novel method named "**Adaptive Multi-teacher Self-distillation (AMS)**", which combines multi-teacher self-distillation method and adaptive weight mechanism to alleviate the forgetting issue in AT. Our initial idea comes from *Knowledge distillation* (Hinton et al., 2015), which is a method that can capture more refined and subtle feature information by aligning the model soft output between old and new tasks. However, the knowledge distillation methods employed in CL (e.g., Lwf (Li & Hoiem, 2017) and iCaRL (Rebuffi et al., 2017)) may not be sufficiently sensitive to certain types of perturbations, leading to suboptimal defense effectiveness in these situations. For example, one "stage" of AT may be sensitive to pixel-level noise, while another may be sensitive to geometric transformations or color shifts. To address this issue, we propose a multi-teacher self-distillation method designed to capture diverse feature representations, thereby enhancing the model's ability to cope with various forms of perturbations. As explained before, another issue is that the information transferred from earlier stages to later stages is not always reliable. So, we design a "**Reweighting-based Loss Correction**" mechanism to adaptively eliminate the negative effects caused by "bad" teachers. Specifically, we utilize the softmax outputs of the teacher models to determine the weight for each sample based on the probability of the true class. Higher probabilities indicate a greater confidence of correct classification, and thus we assign higher weights to these samples and lower weights to those misclassified ones. Overall, this mechanism ensures that misclassified samples do not significantly affect the loss function of distillation module.

We validate the effectiveness of our proposed AMS method through the experiments on a range of datasets (CIFAR-10/100, SVHN, and TinyImageNet) and model architectures (PreActResNet-18 and WideResNet-34/28-10). The experimental results suggest that our method can enhance state-of-the-art adversarial training methods, improving adversarial robustness accuracy across various datasets and architectures.

## 2 BACKGROUND

### 2.1 ADVERSARIAL TRAINING

Let $\mathcal{D} = \{(\mathbf{x}_i, y_i)\}_{i=1}^n$ denote a $C$-classes training dataset with $n$ samples from the original data distribution $\mathcal{P}$, where $\mathbf{x}_i \in \mathbb{R}^d$ is a natural sample in the $d$-dimensional input space and $y_i \in \{1, \ldots, C\}$ is its ground true label. Vanilla adversarial training can be formulated as a *minimax robust optimization problem* (Madry et al., 2018):

$$\min_{\boldsymbol{\theta}} \mathbb{E}_{\mathbf{x}_i \sim \mathcal{P}} \left[ \max_{\mathbf{x}_i' \in \mathcal{B}_p(\mathbf{x}_i, \epsilon)} \mathcal{L}(h_{\boldsymbol{\theta}}(\mathbf{x}_i'), y_i) \right], \tag{1}$$

where $h_{\boldsymbol{\theta}} : \mathbb{R}^d \to \mathbb{R}^C$ is a DNN classifier with parameter $\boldsymbol{\theta}$, $(\mathbf{x}_i', y_i)$ is an adversarial example of $(\mathbf{x}_i, y_i)$, and $\mathcal{B}_p(\mathbf{x}_i, \epsilon) = \{\mathbf{x}_i' : \|\mathbf{x}_i' - \mathbf{x}_i\|_p \leq \epsilon\}$ is an adversarial region centered at $\mathbf{x}_i$ with radius $\epsilon > 0$ under the $\ell_p$ norm-bounded perturbation (e.g., $\ell_\infty$ (Rice et al., 2020) and $\ell_2$ (Carlini et al., 2019) norms), $\mathcal{L}(h_{\boldsymbol{\theta}}(\mathbf{x}_i'), y_i)$ is the cross-entropy loss function on the adversarial example $(\mathbf{x}_i', y_i)$. As explained in (Madry et al., 2018), the inner maximization problem is to find the worst-case samples for the given model, while the outer minimization problem aims to train a model robust to adversarial examples. To solve this problem, Madry et al. (2018) employed a multi-step "Projected

Gradient Descent (PGD)" method, which starts at a randomly initialized point $\mathbf{x}^0$ in $\mathcal{B}_p(\mathbf{x}_i, \epsilon)$ and iteratively updates the adversarial example by

$$\mathbf{x}'_i = \Pi_{\mathcal{B}_p(\mathbf{x}_i, \epsilon)} \left( \mathbf{x}'_i + \alpha \cdot \text{sign} \left( \nabla_{\mathbf{x}_i} \mathcal{L} \left( h_{\boldsymbol{\theta}}(\mathbf{x}'_i), y_i \right) \right) \right), \quad (2)$$

where $\alpha$ is the step size, $\Pi(\cdot)$ is the projection operator. We refer to this inner maximization problem with $K$ steps as "PGD-K". Numerous studies have been proposed based on PGD, including (Jia et al., 2022; Kannan et al., 2018; Chen et al., 2022; Rice et al., 2020).

Besides PGD, another representative AT method is TRADES (Zhang et al., 2019b), which balances the trade-off between robust and clean accuracy by minimizing a different adversarial loss

$$\min_{\boldsymbol{\theta}} \mathbb{E}_{\mathbf{x}_i \sim \mathcal{P}} \left\{ \mathcal{L}(h_{\boldsymbol{\theta}}(\mathbf{x}_i), y_i) + \beta \cdot \max_{\mathbf{x}'_i \in \mathcal{B}_p(\mathbf{x}_i, \epsilon)} D_{\text{KL}}(h_{\boldsymbol{\theta}}(\mathbf{x}_i) \parallel h_{\boldsymbol{\theta}}(\mathbf{x}'_i)) \right\}, \quad (3)$$

where $D_{\text{KL}}(\cdot \parallel \cdot)$ is the Kullback-Leibler divergence. The first term in Eq.(3) contributes to clean accuracy, and the second term with hyper-parameter $\beta$ (set $\beta = 6$ as default) can be seen as a regularization for adversarial robustness that balances the outputs of clean and adversarial examples. Furthermore, a substantial body of related work based on TRADES has been proposed, including (Wu et al., 2020; Cui et al., 2021; Jin et al., 2023).

In addition to PGD-based and TRADES-based methods, several enhancements for AT have been made by employing other strategies, including unsupervised/self-supervised learning (Alayrac et al., 2019; Chen et al., 2020b;a; Naseer et al., 2020), data augmentation (Lee et al., 2020; Rebuffi et al., 2021b; Gowal et al., 2021; Rice et al., 2020; Wu et al., 2020), and generative model (Dong et al., 2020; Pang et al., 2022; Wang et al., 2023). Furthermore, due to the high computational cost of AT, various efforts have been made to accelerate the training process. These include reusing computations (Shafahi et al., 2019; Zhang et al., 2019a) and adopting one-step training methods (Wong et al., 2020; Jia et al., 2024; Vivek & Babu, 2020).

## 2.2 CONTINUAL LEARNING AND KNOWLEDGE DISTILLATION

Continual learning (CL) (Wang et al., 2024; Lopez-Paz & Ranzato, 2017), a rapidly developing field in deep learning, aims to develop systems capable of learning continuously from sequential or streaming data while retaining previously acquired knowledge. A significant challenge that CL encounters is "catastrophic forgetting" (McClelland et al., 1995; McCloskey & Cohen, 1989), where DNNs forget old knowledge upon learning new information. Various strategies have been proposed to mitigate this phenomenon in CL, primarily focusing on regularization-based methods and replay-based methods (De Lange et al., 2021). Roughly speaking, the replay-based methods method utilizes reservoir sampling (Vitter, 1985) to maintain historical data (e.g., ER (Chaudhry et al., 2019), GCR (Tiwari et al., 2022) and DGC-ER (Lin et al., 2024)) or logits (e.g., DER (Buzzega et al., 2020)) in the memory buffer, then extract new incoming training data with random samplings for learning the current task. Another way to solve continual learning is through deliberately designed regularizer terms. For example, the method EWC (Kirkpatrick et al., 2017) adds a quadratic penalty in the loss function, which penalizes the variation of each parameter depending on its contribution to old tasks; Lwf (Li & Hoiem, 2017) learns new training samples while using their predictions from the output head of the old tasks to compute the distillation loss.

Knowledge distillation (KD) (Hinton et al., 2015) is a method that distills knowledge from a larger deep neural network into a small network (Li et al., 2020; Polino et al., 2018; Zhang et al., 2018a;b). KD learning schemes (Gou et al., 2021) can be divided into three main categories based on whether the teacher model is updated simultaneously with the student model or not: offline distillation (Hinton et al., 2015; Mirzadeh et al., 2020), online distillation (Zhang et al., 2018b; Wu & Gong, 2021), and self-distillation (Zhang et al., 2019c; Hou et al., 2019; Yang et al., 2019). In self-distillation, the same network serves as both the teacher and the student models, enabling direct application in CL scenarios, e.g., iCaRL (Rebuffi et al., 2017) and Lwf (Li & Hoiem, 2017).

## 3 CATASTROPHIC FORGETTING IN AT

In this section, we first explain that why AT can be regarded as a "CL-style" problem, from the perspective of data distribution shifts during the training process. Then, we investigate the phenomenon of forgetting in AT through an empirical study.

**Regard AT as a CL-style problem.** We are aware that AT involves the model adapting to continually changing distributions of adversarial samples, and we will explain that it shares some similar

properties with a CL scenario that adapts the model to new tasks or changing environments. Recall from Eq.(1) that vanilla adversarial training is formulated as a minimax robust optimization problem, utilizing the original data distribution $\mathcal{P}$ and a DNNs classifier $h_\theta$. In each epoch of the training process, the parameter is updated by a set of adversarial samples, denoted by $\mathcal{P}^{adv}$, based on the inner maximization problem, where $\mathcal{P}^{adv}$ can be viewed as a sample set selected from an implicit distribution by adding adversarial perturbations to the original distribution. To distinguish the different $\mathcal{P}^{adv}$s at different epochs, we add a subscript to it; for example, $\mathcal{P}^{adv}_t$ denotes the sample $\mathcal{P}^{adv}$ at the $t$-th epoch. The knowledge required to solve the outer minimization problem (i.e., minimizing the loss $\mathcal{L}$ on $\mathcal{P}^{adv}_t$) is different from the knowledge needed for optimizing on $\mathcal{P}^{adv}_{t-1}$. In other words, the tasks training on $\mathcal{P}^{adv}_{t-1}$ and $\mathcal{P}^{adv}_t$ are in fact different. The sequence of changing adversarial samples $\{\mathcal{P}^{adv}_t\}_{t=1}^T$ induce a sequence of tasks for the classifier $h_\theta$. Since the inner maximization optimal problem at epoch $t$ can only generate the sample $\mathcal{P}^{adv}_t$, the model at this current epoch cannot revisit the knowledge from previous tasks, which is similar to a CL scenario.

**Does catastrophic forgetting exist in AT?** To answer this question, we further conduct an empirical study to verify the forgetting phenomenon in AT. We first define "**stage**" and "**stage sequence**" within the context of AT, which are similar to "task" and "task sequence" in CL. Let $\mathcal{P}^{adv}_t$ represent the adversarial samples generated in the $t$-th epoch. Suppose there are $T$ epochs in total. We divide the training process into multiple stages with each stage containing $m$ epochs:

$$\{\underbrace{\mathcal{P}^{adv}_1,\ldots,\mathcal{P}^{adv}_m}_{\mathcal{S}_1},\underbrace{\mathcal{P}^{adv}_{m+1},\ldots,\mathcal{P}^{adv}_{2m}}_{\mathcal{S}_2},\ldots,\underbrace{\mathcal{P}^{adv}_{(\lceil\frac{T}{m}\rceil-1)\times m+1},\ldots,\mathcal{P}^{adv}_T}_{\mathcal{S}_{\lceil\frac{T}{m}\rceil}}\}.$$

Thus, we define the stage sequence as $\mathcal{S}=\{\mathcal{S}_1,\ldots,\mathcal{S}_{\lceil\frac{T}{m}\rceil}\}$. The experiment is designed as follows. We first train a model by an AT method and collect the adversarial samples. Specifically, we collect correctly classified adversarial samples in the $s$-th stage and form a new "correctly classified adversarial dataset", which is denoted by $\mathcal{C}^{adv}_s$. This process continues until the end, producing a collection of correctly classified adversarial dataset denoted as $\mathcal{C}^{adv}=\{\mathcal{C}^{adv}_1,\ldots,\mathcal{C}^{adv}_{\lceil\frac{T}{m}\rceil}\}$. **After completing the training process**, we evaluate the classification accuracy on each collected dataset $\mathcal{C}^{adv}_s$. Meanwhile, we employ the *Final Average Accuracy* (FAA) (Douillard et al., 2020; Hou et al., 2019) and *Final Forgetting* (FF) (Chaudhry et al., 2019) to assess the forgetting degree. These two metrics are both widely used for continual learning.

Table 1: Verification results of the forgetting phenomenon in adversarial training across different datasets and perturbation threat models. The total epochs are evenly divided into 5 stages. Note that the "adversarial datasets" collected at each stage are initially classified correctly, resulting in **an initial accuracy of 100%** for each dataset. Let $a_{k,j}$ (for $k \geq j$) denote the classification accuracy evaluated on the testing set of the stage $j$ after learning stage $k$. The FAA is defined as FAA $\triangleq \frac{1}{T}\sum_{j=1}^T a_{T,j}$; the FF is defined as FF $\triangleq \frac{1}{T-1}\sum_{j=1}^{T-1}(max_{k\in\{1,\ldots,T-1\}}a_{k,j}-a_{T,j})$. These two metrics quantify the degree of forgetting, where higher FAA and lower FF indicate less forgetting of previously learned knowledge.

| Dataset | Norm | Robust Test Accuracy (%) | | | | | FAA (%) | FF (%) |
| --- | --- | --- | --- | --- | --- | --- | --- | --- |
| | | stage 1 | stage 2 | stage 3 | stage 4 | stage 5 | | |
| CIFAR-10 | $\ell_\infty$ | 88.36 | 91.43 | 89.21 | 90.65 | 90.21 | 89.97 | 10.03 |
| | $\ell_2$ | 89.34 | 90.67 | 90.56 | 91.84 | 90.63 | 90.61 | 9.39 |
| CIFAR-100 | $\ell_\infty$ | 77.74 | 87.87 | 88.33 | 89.25 | 89.35 | 86.51 | 13.49 |
| | $\ell_2$ | 78.89 | 89.34 | 88.21 | 88.50 | 90.17 | 87.02 | 12.98 |
| SVHN | $\ell_\infty$ | 88.61 | 92.52 | 92.45 | 92.91 | 90.18 | 91.33 | 8.67 |
| | $\ell_2$ | 86.72 | 92.66 | 90.57 | 90.47 | 91.81 | 90.45 | 9.55 |
| TinyImageNet | $\ell_\infty$ | 58.98 | 81.53 | 80.70 | 81.05 | 75.83 | 75.62 | 24.38 |
| | $\ell_2$ | 65.18 | 81.85 | 79.70 | 82.37 | 82.73 | 78.37 | 22.63 |

In Table 1, we observe that forgetting occurs across a variety of datasets (e.g., SVHN, CIFAR-10, CIFAR-100 and TinyImageNet), perturbation threat models (e.g., $\ell_2$ and $\ell_\infty$) and different stages of training, indicating that it might be a general property in adversarial training. We can find a significant gap between the robust test performances at the stage when the datasets are collected

and the final robust test performance at the end of training. Notably, under the $\ell_\infty$ threat model, FAA decreased significantly from 100% to 86.51% for CIFAR-100 and from 100% to 75.62% for TinyImageNet. We also investigate the phenomenon of forgetting in adversarial training via some other supplemental experiments in Appendix B.1 and B.2.

# 4 OUR METHODOLOGY

For ease of understanding, we briefly introduce our high-level idea below, and then elaborate on each technical part.

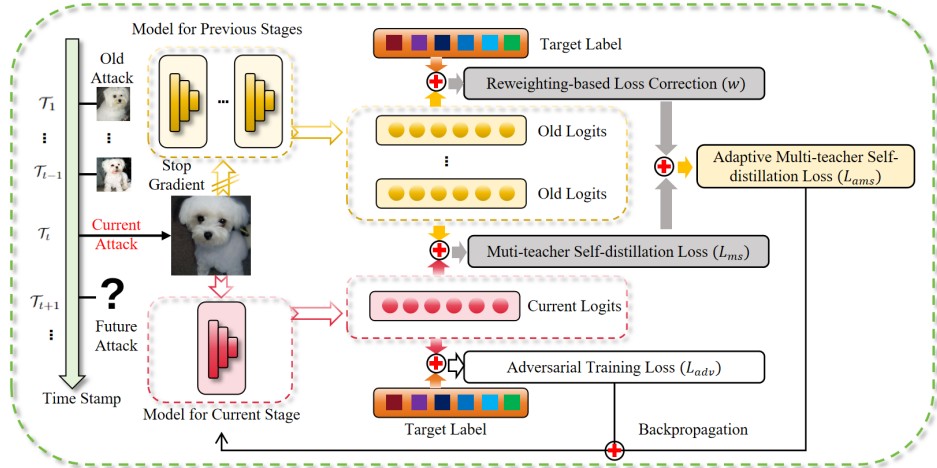

Figure 2: Framework of our AMS. The upper module (in yellow block) consists of the model for previous stages, with the aim of maintaining knowledge of previous stages. The lower module (in red block) is the vanilla adversarial training process.

**Overview.** Our goal is to design a method to enhance robust accuracy through alleviating forgetting. We consider using multi-teacher self-distillation technology, which combines multi-teacher distillation architecture and self-distillation algorithm to transfer the subtle feature information learned from previous "stages". However, since adversarial training models usually do not have high classification accuracy (say, $\geq 90\%$), which may lead to transferring more previous incorrect information to the subsequent "stages". To alleviate the negative effects caused by the incorrect information from previous "stages", we propose an adaptive sample reweighting mechanism called "Reweighting-based Loss Correction" to correct the loss function in distillation module. The overall framework of our method can be seen in Figure 2 and the complete algorithm is presented in Algorithm 1.

**Multi-teacher Self-distillation.** A direct way to employ multi-teacher self-distillation technique is to treat each training epoch in the adversarial training process as a separate "stage". However, since the adversarial training process requires a large number of epochs (e.g., 200 as in (Madry et al., 2018)), directly applying this method could result in a rapid increase in both training time and memory as training progresses. So we take every $m$ epochs of the training process as one stage, as illustrated in Section 3 before.

Now we integrate the multi-teacher self-distillation approach into the PGD framework to transfer knowledge learned from previous stages, thereby mitigating forgetting and improving the model's robust accuracy. Generally, assuming we have a model $h_{\boldsymbol{\theta}_t}$ parameterized by $\boldsymbol{\theta}_t$ at epoch $t$ within stage $s$, the multi-teacher self-distillation approach can be formulated as:

$$\mathcal{L}_{ms} = \sum_{j=1}^{s-1} D_{\mathrm{KL}}(h_{\boldsymbol{\theta}_j}(\mathbf{x}'_i) \parallel h_{\boldsymbol{\theta}_t}(\mathbf{x}'_i)), \tag{4}$$

where $D_{\mathrm{KL}}(\cdot \parallel \cdot)$ is the Kullback-Leibler divergence. $\{\boldsymbol{\theta}_j\}_{j=1}^{s-1}$ represent the model parameters from all stages before stage $s$, and $\mathbf{x}'_i$ is an adversarial sample in epoch $t$. Thus, the training objective of AT with multi-teacher self-distillation can be expressed as

$$\mathcal{L} = \mathcal{L}_{adv} + \frac{\lambda}{s-1} \cdot \mathcal{L}_{ms}, \tag{5}$$

---

**Algorithm 1** Adversarial training with AMS

---

**Input:** Training datasets $\mathcal{D} = (\mathbf{x}_i, y_i)_{i=1}^n$, perturbation bound $\epsilon$, learning rate $\tau$, step size $\alpha$, number of iterations $K$ in inner optimization, network architecture parameterized by $\boldsymbol{\theta}$, memory buffer $\mathcal{M}$, regularization parameter $\lambda$, interval parameter $m$, and the number of epoch $T$.

**Output:** Robust network with parameter $\boldsymbol{\theta}_T$

1: Initialize $\boldsymbol{\theta}$
2: $s \leftarrow 0, \mathcal{M} \leftarrow \{\}$
3: **for** $t = 1, \ldots, T$ **do**
4:     Sample $\mathbf{x}_i$ from $\mathcal{D}$
5:     **for** $i = 1, \ldots, n$ **do**
6:         $\mathbf{x}_i^0 \leftarrow \mathbf{x}_i + 0.001 \cdot \mathcal{N}(\mathbf{0}, \mathbf{I})$
7:         **for** $k = 1, \ldots, K$ **do**
8:             $\mathbf{x}_i' = \Pi_{\mathcal{B}_p(\mathbf{x}_i, \epsilon)} \left( \mathbf{x}_i' + \alpha \cdot \text{sign} \left( \nabla_{\mathbf{x}_i} \mathcal{L} \left( h_{\boldsymbol{\theta}}(\mathbf{x}_i'), y_i \right) \right) \right)$
9:         **end for**
10:        **for** $\boldsymbol{\theta}_j \in \mathcal{M}$ **do**
11:           $w_{i,j} = \left[ \text{softmax}(h_{\boldsymbol{\theta}_j}(\mathbf{x}_i')) \right]_{y_i}$ */ Reweightng-based Loss Correction */
12:        **end for**
13:        $\boldsymbol{\theta}_{i+1} \leftarrow \boldsymbol{\theta}_i - \tau \cdot \frac{1}{n} \sum_{i=1}^n \nabla(\mathcal{L}_{adv} + \frac{\lambda}{\sum_{j=1}^{s-1} w_{i,j}} \cdot \sum_{j=1}^{s-1} w_{i,j} \cdot D_{\text{KL}}(h_{\boldsymbol{\theta}_j}(\mathbf{x}_i') \parallel h_{\boldsymbol{\theta}_t}(\mathbf{x}_i')))$
14:        */* Adaptive Multi-teacher Self-distillation */*
15:     **end for**
16:     **if** $\text{mod}(t, m) == 0$ **then**
17:        $\mathcal{M} \leftarrow \mathcal{M} \cup \{\boldsymbol{\theta}_t\}$ */ Save model parameters */
18:        $s \leftarrow s + 1$
19:     **end if**
20: **end for**

---

where the regularization parameter $\lambda$ controls the *plasticity-stability* trade-off. Too much focus on stability will hinder the model's ability to learn new adversarial samples, whereas excessive plasticity will lead to greater forgetting of previously learned knowledge. The fraction $\frac{1}{s-1}$ is used to calculate the average of all distillation losses. The term "$\mathcal{L}_{adv}$" is the adversarial loss of current epoch $t$ and can be formulated as

$$\mathcal{L}_{adv} = \max_{\mathbf{x}_i' \in \mathcal{B}_p(\mathbf{x}_i, \epsilon)} \mathcal{L}(h_{\boldsymbol{\theta}_t}(\mathbf{x}_i'), y_i). \tag{6}$$

Ideally, we look for parameters that fit the current stage well while retaining the correct knowledge learned in the previous stages, thereby enhancing the model's ability to cope with various forms of perturbations. However, during knowledge distillation, a large amount of incorrect information made by previous stage models can be transferred to current stage model along with parameters. This occurs due to the low robust accuracy of AT models. For example, the best robust accuracy on CIFAR-10 is less than 60%.

**Reweightng-based Loss Correction (RLC).** To resolve the above "transferring-incorrect-information" problem, we propose a Reweighting-based Loss Correction mechanism to adaptively eliminate the negative effects caused by "bad" teachers. Specifically, for a teacher model $h_{\boldsymbol{\theta}_j}$, we first use the softmax function to compute the probability output of adversarial samples that are generated in the current stage. Then, we extract the probability corresponding to the true class for each adversarial sample, denoted as $w_{\cdot,j}$. This probability $w_{\cdot,j}$ quantifies the degree of correct information that can be transferred from the teacher model. Therefore, for each adversarial sample $\mathbf{x}'$, we employ $w_{\cdot,j}$ as a distillation weight to correct the loss function $\mathcal{L}_{ms}$ (see Eq.(4)). Mathematically, the probability of an adversarial sample $\mathbf{x}_i'$ with true label $y_i$ can be expressed as:

$$w_{i,j} = \left[ \text{softmax}(h_{\boldsymbol{\theta}_j}(\mathbf{x}_i')) \right]_{y_i}. \tag{7}$$

As a consequence, we modify our approach to be an "adaptive" way, where the loss function of Eq.(4) is changed to be:

$$\mathcal{L}_{ams} = \frac{\lambda}{\sum_{j=1}^{s-1} w_{i,j}} \cdot \sum_{j=1}^{s-1} w_{i,j} \cdot D_{\text{KL}}(h_{\boldsymbol{\theta}_j}(\mathbf{x}_i') \parallel h_{\boldsymbol{\theta}_t}(\mathbf{x}_i')). \tag{8}$$

## 5 EXPERIMENTS

In this section, we first discuss the hyper-parameter sensitivity of our method, detailed in Section 5.1. We then evaluate the robustness on benchmark datasets in Section 5.2. Section 5.3 presents our findings that our method can mitigate the robust overfitting issue to a certain degree. Furthermore, in Section 5.4, we compare our approach with various CL methods. Additional experimental results are explained in Section 5.5. Finally, we conduct the ablation study in Section 5.6.

**Experimental Setting.** We default to using PreActResNet-18 (He et al., 2016) for $\ell_2$ and $\ell_\infty$ threat models on CIFAR-10/100 (Krizhevsky, 2009), TinyImageNet and SVHN (Netzer et al., 2011). In addition, we also use WideResNet-34-10 (Zagoruyko & Komodakis, 2016) for CIFAR-10/100 with $\ell_\infty$ threat model (for a proper comparison to the results reported for TRADES (Zhang et al., 2019b)), and use WideResNet-28-10 for TinyImageNet (TinyImagenet dataset is a subset of ImageNet (Deng et al., 2009)). For CIFAR-10/100, we train with the SGD optimizer with Nesterov momentum (Nesterov, 1983) 0.9 using a batch size of 128, a step-wise learning rate initially at 0.1, and weight decay $5 \times 10^{-4}$. For SVHN, we use the same parameters except with a starting learning rate of 0.01 instead. For TinyImageNet, we use the cyclic learning rate schedule with cosine annealing (Smith & Topin, 2019), where the initial learning rate is set to 0.2. We adopt the widely used adversarial training setting (Madry et al., 2018; Zhang et al., 2019b). For the $\ell_\infty$ threat model, we using radius $\epsilon = 8/255$ and step size $\alpha = 2/255$; For the $\ell_2$ threat model, we using radius $\epsilon = 128/255$ and step size $\alpha = 15/255$. For vanilla adversarial training, the training examples are generated with 10 steps. We also adopt the widely used data augmentation technology, such as random horizontal flip and $32 \times 32$ random crop with 4-pixel padding. We report the test accuracy on the best checkpoint that achieves the highest robust validation accuracy under PGD-20. We implement and train our model with PyTorch on a 64-bit Linux machine with 8 NVIDIA RTX A6000 Ada GPUs.

**Evaluation Setting.** We evaluate the model robustness against wihte-box attacks, black-box attack and Auto Attack. For white-box attacks, we utilize PGD-20 and CW-20 (Carlini & Wagner, 2017) to evaluate the trained models. For the black-box attacks (Papernot et al., 2017), we generate adversarial examples by attacking a local substitute model trained with the vanilla adversarial training method. These adversarial examples are then applied to the defense model to evaluate its performance. For the black-box attacks, we utilize PGD-20 and CW-20 as the attack methods. As one of the strongest attacks for evaluating model robustness, Auto Attack (Croce & Hein, 2020b) contains an ensemble of diverse attacks, including three wihte-box attacks (APGD-CE (Croce & Hein, 2020b), APGD-DLR (Croce & Hein, 2020b) and FAB (Croce & Hein, 2020a)) and one black-box attack (Square Attack (Andriushchenko et al., 2020)).

### 5.1 SENSITIVITY OF HYPER-PARAMETER

In our proposed Algorithm 1, we need to determine two parameters, the regularization parameter $\lambda$ and the epoch interval parameter $m$. We conduct the numerical experiments to illustrate how these hyper-parameters influence the performance of our algorithm. For the regularization parameter $\lambda$, we train the model with $\lambda \in \{0.05, 0.1, 0.2, 0.3, 0.5, 1.0, 2.0, 3.0\}$. For the interval parameter $m$, we train the model with $m \in \{10, 20, 30, 50\}$.

From Table 4 in Appendix B.3, we observe that as the regularization parameter $\lambda$ increases, both the clean accuracy and robust accuracy initially go up and then down. Taking both metrics into account, we take $\lambda = 0.5$ as a proper choice for our method. Meanwhile, we evaluate the interval parameters $m$ comprehensively from four aspects: memory usage, time consumption, clean accuracy, and robust accuracy, as illustrated in Figure 5 in Appendix B.3. We find that $m = 20$ is an appropriate setting for our approach. So, in the following experiments, we set $\lambda = 0.5$ and $m = 20$ as the default parameters.

### 5.2 COMPARISON WITH SOTA ON WIDERESNET

In Table 2, we compare our method with the state-of-the-art methods on WideResNet across CIFAR-10/100, SVHN and TinyImageNet, under the $\ell_\infty$ norm with $\epsilon = 8/255$. The state-of-the-art methods include AVMixup (Lee et al., 2020), MART (Wang et al., 2019), ES (Rice et al., 2020), FAB (Zhang et al., 2020), LS (Pang et al., 2021), LBGAT (Cui et al., 2021), S²O (Jin et al., 2022), RAT (Jin et al., 2023), AWP (Wu et al., 2020), EMA (Gowal et al., 2020), (Sehwag et al., 2022), (Gowal et al.,

2021), (Rebuffi et al., 2021a) and (Wang et al., 2023). We integrate AMS with TRADES, (Sehwag et al., 2022), and (Wang et al., 2023), and assess their performances in different datasets. The results in Table 2 suggest that our method consistently improves robust accuracy across different datasets and models. Especially for TinyImageNet under Auto Attack, our method achieves a robust test accuracy of 31.52%.

Table 2: Test accuracy (%) of the proposed methods and current state-of-the-art methods on CIFAR-10/100, SVHN and TinyImageNet under the $\ell_\infty$ norm with $\epsilon = 8/255$. We highlight the best results in **bold**. Note that "*" means under PGD-40 attack, and "**" means under PGD-10 attack. The numerical results of the baseline methods are quoted from their papers. "-" indicates that the result is not reported in the original paper. The notations "($\uparrow \Delta$)" and "($\downarrow \Delta$)" indicate percentage increases and decreases compared to the baseline, respectively. The term "1M" denotes the utilization of one million images generated by a class-conditional Denoising Diffusion Probabilistic Model (Ho et al., 2020).

| Dataset | Method | Architecture | Clean | PGD-20 | CW-20 | AA |
|---------|--------|--------------|-------|--------|-------|-----|
| CIFAR-10 | AVMixup (Lee et al., 2020) | WideResNet-34-10 | **92.56** | 59.75 | 54.34 | 39.70 |
| | Gowal et al. (2020) | WideResNet-70-16 | 85.29 | 58.22* | - | 57.20 |
| | MART (Wang et al., 2019) | WideResNet-34-10 | 83.51 | 58.31 | 54.33 | 51.10 |
| | ES (Rice et al., 2020) | WideResNet-34-20 | 85.34 | - | - | 53.42 |
| | FAB (Zhang et al., 2020) | WideResNet-34-10 | 84.52 | - | - | 53.51 |
| | LS (Pang et al., 2021) | WideResNet-34-20 | 86.43 | 57.91** | - | 54.39 |
| | S²O (Jin et al., 2022) | WideResNet-34-20 | 86.01 | 61.12 | 57.93 | 55.90 |
| | RAT (Jin et al., 2023) | WideResNet-34-10 | 85.98 | 58.47 | 56.13 | 54.20 |
| | S²O (Jin et al., 2022) | WideResNet-34-10 | 85.58 | 59.43 | 55.66 | 53.93 |
| | S²O+AMS | WideResNet-34-10 | 85.83 ($\uparrow$ 0.25) | 61.29 ($\uparrow$ 1.86) | 57.78 ($\uparrow$ 2.12) | 55.92 ($\uparrow$ 1.99) |
| | Wang et al. (2023) (1M) | WideResNet-34-10 | 91.18 | 68.11 | 65.20 | 63.31 |
| | Wang et al. (2023) (1M)+AMS | WideResNet-34-10 | 90.79 ($\downarrow$ 0.39) | **69.32** ($\uparrow$ 1.21) | **66.03** ($\uparrow$ 0.83) | **63.97** ($\uparrow$ 0.66) |
| | TRADES (Zhang et al., 2019b) | WideResNet-34-10 | 84.65 | 56.68 | 54.49 | 53.00 |
| | TRADES+**EMA** (Gowal et al., 2020) | WideResNet-34-10 | 84.78 | 57.23 | 55.12 | 53.76 |
| | TRADES+**AMS** | WideResNet-34-10 | 85.37 ($\uparrow$ 0.72) | 58.76 ($\uparrow$ 2.08) | 56.43 ($\uparrow$ 1.94) | 54.31 ($\uparrow$ 1.31) |
| | TRADES+**AWP** (Wu et al., 2020) | WideResNet-34-10 | 84.99 | 59.67 | 57.41 | 56.17 |
| | TRADES+**AWP**+**AMS** | WideResNet-34-10 | 85.21 ($\uparrow$ 0.22) | 61.45 ($\uparrow$ 1.78) | 58.56 ($\uparrow$ 1.14) | 57.36 ($\uparrow$ 1.19) |
| CIFAR-100 | Gowal et al. (2020) | WideResNet-70-16 | 60.86 | 31.47* | - | 30.03 |
| | Rebuffi et al. (2021a) (1M) | WideResNet-28-10 | 62.41 | - | - | 32.06 |
| | LBGAT (Cui et al., 2021) | WideResNet-34-10 | 60.43 | 35.50 | 31.50 | 29.34 |
| | RAT (Jin et al., 2023) | WideResNet-34-10 | 62.93 | 33.36 | 29.61 | 27.90 |
| | TRADES (Zhang et al., 2019b) | WideResNet-34-10 | 60.22 | 32.11 | 28.93 | 26.90 |
| | TRADES+**EMA** (Gowal et al., 2020) | WideResNet-34-10 | 61.43 | 32.76 | 29.41 | 27.19 |
| | TRADES+**AMS** | WideResNet-34-10 | 62.56 ($\uparrow$ 2.34) | 33.81 ($\uparrow$ 1.70) | 30.47 ($\uparrow$ 1.54) | 28.25 ($\uparrow$ 1.35) |
| | Sehwag et al. (2022) (1M) | WideResNet-34-10 | **65.76** | 36.33 | 32.97 | 31.20 |
| | Sehwag et al. (2022) (1M)+**AMS** | WideResNet-34-10 | 65.55 ($\downarrow$ 0.21) | **36.42** ($\uparrow$ 0.09) | **33.56** ($\uparrow$ 0.59) | **31.46** ($\uparrow$ 0.26) |
| SVHN | Gowal et al. (2021) | WideResNet-28-10 | 92.87 | - | - | 56.83 |
| | Gowal et al. (2021) (1M) | WideResNet-28-10 | 94.15 | - | - | 60.90 |
| | Rebuffi et al. (2021a) (1M) | WideResNet-28-10 | 94.39 | - | - | 61.09 |
| | Wang et al. (2023) (1M) | WideResNet-28-10 | 95.08 | 65.38 | 62.98 | 61.73 |
| | Wang et al. (2023) (1M)+**AMS** | WideResNet-28-10 | **95.26** ($\uparrow$ 0.18) | **66.67** ($\uparrow$ 1.29) | **64.21** ($\uparrow$ 1.23) | **63.15** ($\uparrow$ 1.42) |
| TinyImageNet | Gowal et al. (2021) | WideResNet-28-10 | 51.56 | - | - | 21.56 |
| | Gowal et al. (2021) (1M) | WideResNet-28-10 | 60.95 | - | - | 26.66 |
| | Wang et al. (2023) (1M) | WideResNet-28-10 | **64.83** | 32.21 | 31.65 | 30.76 |
| | Wang et al. (2023) (1M)+**AMS** | WideResNet-28-10 | 64.20 ($\downarrow$ 0.63) | **33.46** ($\uparrow$ 1.25) | **32.54** ($\uparrow$ 0.89) | **31.52** ($\uparrow$ 0.76) |

## 5.3 ROBUST OVERFITTING

In our experiments, we also observe an interesting phenomenon that our method can alleviate robust overfitting (Rice et al., 2020), though it is not the main focus in our paper. As a common issue in adversarial training, the property of adversarial robust overfitting is that after the first and second learning rate decay, further training will continue to substantially decrease the robust test accuracy. To illustrate this issue, we show the comparisons on test accuracy with the baselines in Appendix B.4 (Table 6) from both the best checkpoint, which achieves the highest robust test accuracy under PGD-20 (denoted as "Best"), and the final checkpoint (denoted as "Final"), as well as the differential between these two checkpoints (denoted as "Diff"). A larger value of "Diff" indicates a more pronounced occurrence of robust overfitting. We observe that the accuracy difference between the best and final test results from our method is reduced to around $0.5\%$, while the gaps for TRADES are significantly larger, at 3.28% under the PGD-20 attack on CIFAR-10. This indicates that our method has the potential to mitigate the robust overfitting issue. Meanwhile, we also compared with some related works that mitigate robust overfitting in Appendix B.4 (Table 7).

## 5.4 Comparison with Different CL Methods in Appendix B.5

In Table 8 and Table 9, we compare our method against various CL methods, such as replay-based methods (e.g., ER (Chaudhry et al., 2019), DER (Buzzega et al., 2020), DGC-ER (Lin et al., 2024), GCR (Tiwari et al., 2022)), regularization-based methods (e.g., EWC (Kirkpatrick et al., 2017), Lwf (Li & Hoiem, 2017)). The experimental parameter settings for these CL methods are provided in the Appendix B.5. From the results in Table 8, we observe that almost all the CL methods achieve certain extent improvements on robust accuracy. For example, Lwf (Li & Hoiem, 2017) has the increased best test robust accuracy from 47.70% to 50.45% under Auto Attack. Compared to other CL methods, our approach has larger improvement in robust accuracy, while maintaining clean accuracy without decline; for example, the best test robust accuracy increases from 47.70% to 51.63% (+3.93%) under Auto Attack.

## 5.5 Other empirical results in Appendix B.6

We also compare our method with PGD (Madry et al., 2018) using PreActResNet-18 on CIFAR-10/100 and SVHN, under the $\ell_2$ norm with $\epsilon = 128/255$, as detailed in Table 10. Additionally, we evaluate the performance of our method under black-box attacks in Table 11. These results also indicate that our method can enhance robust accuracy to a certain extent, while maintaining clean accuracy without any decrease. Furthermore, as illustrated in Table 12, we have empirically validated the effectiveness of our method in mitigating forgetting. Specifically, compared to the validation results of the vanilla AT method presented in Table 1, the results of our approach reveal significant improvements in FAA and reductions in FF. Finally, we apply AMS to other structures (VGG16 (Simonyan, 2014) and ViT (Dosovitskiy et al., 2021)) with normal adversarial training; the results shown in Table 13 illustrate that our AMS method is effective on these architectures as well.

## 5.6 Ablation experiment.

Given that our proposed method incorporates a "Reweighting-based Loss Correction" (RLC) technique, a natural question arises: how effective is it in improving the final performance? To address this question, we conducted an ablation experiment, and the results are shown in Table 3. The results demonstrate that the use of RLC can enhance robust accuracy to a certain extent. For example, when RLC is removed, the robust accuracy on CIFAR-100 decreases from 28.05% to 27.62% with TRADES+AMS, and from 28.31% to 27.58% with PGD-10+AMS, highlighting the contribution of the RLC component to improving adversarial robustness.

Table 3: Test accuracy (%) of TRADES+AMS and PGD-10+AMS with or without (w/o) the "RLC" term under the $\ell_\infty$ norm with $\epsilon = 8/255$ based on the PreActResNet-18 architecture.

| Method | CIFAR-10 (clean) | CIFAR-10 (AA) | CIFAR-100 (clean) | CIFAR-100 (AA) |
|---|---|---|---|---|
| TRADES+AMS | 84.86 | 51.22 | 60.03 | 28.05 |
| TRADES+AMS w/o RLC | 84.72 | 50.08 | 60.11 | 27.62 |
| PGD-10+AMS | 84.45 | 51.63 | 58.87 | 28.31 |
| PGD-10+AMS w/o RLC | 85.66 | 50.31 | 58.83 | 27.58 |

## 6 Conclusion and Outlook

In this paper, we address the problem of AT from a novel perspective by linking it to the forgetting phenomenon in CL. We first demonstrate that the phenomenon of forgetting indeed occurs in AT. Following this observation, we propose a novel method called Adaptive Multi-teacher Self-distillation (AMS), which employs a carefully designed adaptive regularizer to mitigate forgetting by aligning model outputs between successive stages. Our approach can be seamlessly integrated with several existing AT methods, leading to substantial improvements in robust accuracy. In addition, our experimental results indicate that our method enjoys another benefit, where it can significantly alleviates the issue of robust overfitting for AT. So we conjecture that robust overfitting might be partly caused by the forgetting issue. As the future work, we think it is deserved to explore the connection between forgetting and robust overfitting in greater depth, from both the theoretical and empirical aspects.

ACKNOWLEDGMENTS

The authors would like to thank the anonymous reviewers for their valuable comments and suggestions. This work was partially supported by the National Natural Science Foundation of China (No. 62272432, No. 62432016), the National Key Research and Development Program of China (No. 2021YFA1000900), and the Natural Science Foundation of Anhui Province (No. 2208085MF163).

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

# A    ALGORITHM

In this section, we provide the vanilla AT algorithm (Madry et al., 2018; Zhang et al., 2019b) in Subsection A.1 Algorithm 2. We also summarize the continual learning combine adversarial training implementations, such as replay-based method (i.e., DER (Buzzega et al., 2020)), regularization-based method (i.e., Lwf (Li & Hoiem, 2017)) in Subsection A.2 and Subsection A.3, respectively.

## A.1    VANILLA AT ALGORITHM

In the following, we provide the vanilla adversarial training Algorithm 2.

---

**Algorithm 2** Vanilla Adversarial Training

---

**Input:** Training datasets $\mathcal{D} = (\mathbf{x}_i, y_i)_{i=1}^n$, perturbation bound $\epsilon$, learning rate $\tau$, step size $\alpha$, number of epochs $K$ in inner optimization, network architecture parameterized by $\boldsymbol{\theta}$

**Output:** Robust network with parameter $\boldsymbol{\theta}_T$

1: Initialize $\boldsymbol{\theta}$
2: **for** $t = 1, \ldots, T$ **do**
3:     Sample $\mathbf{x}_i$ from $\mathcal{D}$
4:     **for** $i = 1, \ldots, n$ **do**
5:         $\mathbf{x}_i^0 \leftarrow \mathbf{x}_i + 0.001 \cdot \mathcal{N}(\mathbf{0}, \mathbf{I})$
6:         **for** $k = 1, \ldots, K$ **do**
7:             $\mathbf{x}_i' = \Pi_{\mathcal{B}_p(\mathbf{x}_i, \epsilon)} \left( \mathbf{x}_i' + \alpha \cdot \text{sign} \left( \nabla_{\mathbf{x}_i} \mathcal{L} \left( h_{\boldsymbol{\theta}}(\mathbf{x}_i'), y_i \right) \right) \right)$
8:         **end for**
9:         $\boldsymbol{\theta}_{i+1} \leftarrow \boldsymbol{\theta}_i - \tau \cdot \frac{1}{n} \sum_{i=1}^n \nabla \mathcal{L}_{adv}$
10:     **end for**
11: **end for**

---

## A.2    ADVERSARIAL TRAINING WITH DER ALGORITHM

**Adversarial training with DER.** Dark Experience Replay (DER) is a replay-based continual learning method that aligns the network's logits sampled throughout the optimization trajectory and stored in a memory buffer, thereby promoting consistency with its previous behavior. Specifically, the algorithm employs the Reservoir Sampling strategy (Vitter, 1985) for buffer insertion. Below, we present Algorithm 3, which integrates DER with AT.

## A.3    ADVERSARIAL TRAINING WITH LWF ALGORITHM

**Adversarial training with Lwf.** Learning without Forgetting (Lwf) is a regularization-based continual learning method that trains the network exclusively on new task data while preserving its original capabilities. Below, we present Algorithm 4, which integrates Lwf with AT.

---

**Algorithm 3** Adversarial training with DER

---

**Input:** Training datasets $\mathcal{D} = (\mathbf{x}_i, y_i)_{i=1}^n$, perturbation bound $\epsilon$, learning rate $\tau$, step size $\alpha$, number of epochs $K$ in inner optimization, network architecture parameterized by $\boldsymbol{\theta}$, memory buffer $\mathcal{M}$, regularization parameter $\lambda$

**Output:** Robust network with parameter $\boldsymbol{\theta}_T$

1: Initialize $\boldsymbol{\theta}$
2: $\mathcal{M} \leftarrow \{\}$
3: **for** $t = 1, \ldots, T$ **do**
4:     Sample $\mathbf{x}_i$ from $\mathcal{D}$
5:     **for** $i = 1, \ldots, n$ **do**
6:         $\mathbf{x}_i^0 \leftarrow \mathbf{x}_i + 0.001 \cdot \mathcal{N}(\mathbf{0}, \mathbf{I})$
7:         **for** $k = 1, \ldots, K$ **do**
8:             $\mathbf{x}_i' = \Pi_{\mathcal{B}_p(\mathbf{x}_i, \epsilon)} \left( \mathbf{x}_i' + \alpha \cdot \text{sign} \left( \nabla_{\mathbf{x}_i} \mathcal{L} \left( h_{\boldsymbol{\theta}}(\mathbf{x}_i'), y_i \right) \right) \right)$
9:         **end for**
10:       $(\mathbf{x}_j', \mathbf{z}_j, y_j) \leftarrow sample(\mathcal{M})$
11:       $\mathbf{z}_i \leftarrow h_{\boldsymbol{\theta}}(\mathbf{x}_i^K)$
12:       $reg \leftarrow \lambda \|\mathbf{z}_j' - h_{\boldsymbol{\theta}}(\mathbf{x}_j')\|$
13:       $\boldsymbol{\theta}_{i+1} \leftarrow \boldsymbol{\theta}_i - \tau \cdot \frac{1}{n} \sum_{i=1}^n \nabla[\mathcal{L}_{adv} + reg]$
14:       $\mathcal{M} \leftarrow reservior(\mathcal{M}, (\mathbf{x}_i', \mathbf{z}_i, y))$
15:     **end for**
16: **end for**

---

**Algorithm 4** Adversarial training with Lwf

---

**Input:** Training datasets $\mathcal{D} = (\mathbf{x}_i, y_i)_{i=1}^n$, perturbation bound $\epsilon$, learning rate $\tau$, step size $\alpha$, number of epochs $K$ in inner optimization, network architecture parameterized by $\boldsymbol{\theta}$, regularization parameter $\lambda$

**Output:** Robust network with parameter $\boldsymbol{\theta}_T$

1: Initialize $\boldsymbol{\theta}$
2: **for** $t = 1, \ldots, T$ **do**
3:     Sample $\mathbf{x}_i$ from $\mathcal{D}$
4:     **for** $i = 1, \ldots, n$ **do**
5:         $\mathbf{x}_i^0 \leftarrow \mathbf{x}_i + 0.001 \cdot \mathcal{N}(\mathbf{0}, \mathbf{I})$
6:         **for** $k = 1, \ldots, K$ **do**
7:             $\mathbf{x}_i^k = \Pi_{\mathcal{B}_p(\mathbf{x}_i, \epsilon)} \left( \mathbf{x}_i^{k-1} + \alpha \cdot \text{sign} \left( \nabla_{\mathbf{x}_i} \mathcal{L} \left( h_{\boldsymbol{\theta}}(\mathbf{x}_i^{k-1}), y_i \right) \right) \right)$
8:         **end for**
9:       $\mathcal{L}_{lwf} = D_{\text{KL}}(h_{\boldsymbol{\theta}_{t-1}}(\mathbf{x}_i') \,\|\, h_{\boldsymbol{\theta}_t}(\mathbf{x}_i'))$
10:       $\boldsymbol{\theta}_{i+1} \leftarrow \boldsymbol{\theta}_i - \tau \cdot \frac{1}{n} \sum_{i=1}^n \nabla[\mathcal{L}_{adv} + \lambda \cdot \mathcal{L}_{lwf}]$
11:     **end for**
12: **end for**

---

# B  FULL EXPERIMENTS

In this section, we first introduce another method to verify catastrophic forgetting in adversarial training (AT) in Subsection B.1. We then employ t-SNE analysis (Van der Maaten & Hinton, 2008) to visually demonstrate the dynamic nature of adversarial sample distributions during training in Subsection B.2. Additionally, we present the experimental results for the sensitivity of hyperparameters in Subsection B.3, for robust overfitting in Subsection B.4, and for continual learning (CL) in Subsection B.5. Finally, we provide other empirical details and results in Subsection B.6.

## B.1  ANOTHER METHOD TO VERIFY CATASTROPHIC FORGETTING

In this part, we introduce another method to verify the phenomenon of forgetting in adversarial training. The specific experimental process is as follows:

- **Step 1.** We train a model by an AT method (e.g., PGD or TRADES) on benchmark dataset. We divide the training process into $S$ stages with each stage containing $m$ epochs. In the final epoch of the $s$-th stage, we collect all adversarial samples and form a new "adversarial dataset", which is denoted by $\mathcal{D}_s^{adv}$. This process continues until the end of the training period, resulting in a collection of adversarial datasets denoted as $\mathcal{D}^{adv} = \{\mathcal{D}_1^{adv}, \dots, \mathcal{D}_S^{adv}\}$.

- **Step 2.** After collecting the adversarial dataset, we initialize the AT model. Then, train the initialized model to convergence on each $\mathcal{D}_s^{adv}$ in sequence. Importantly, we do not include samples from $\mathcal{D}_{<s}^{adv}$ while training on $\mathcal{D}_s^{adv}$. After training on dataset $\mathcal{D}_s^{adv}$, we evaluate the accuracy of generated examples that are correctly classified by the classifier on $\mathcal{D}_{\leq s}^{adv}$.

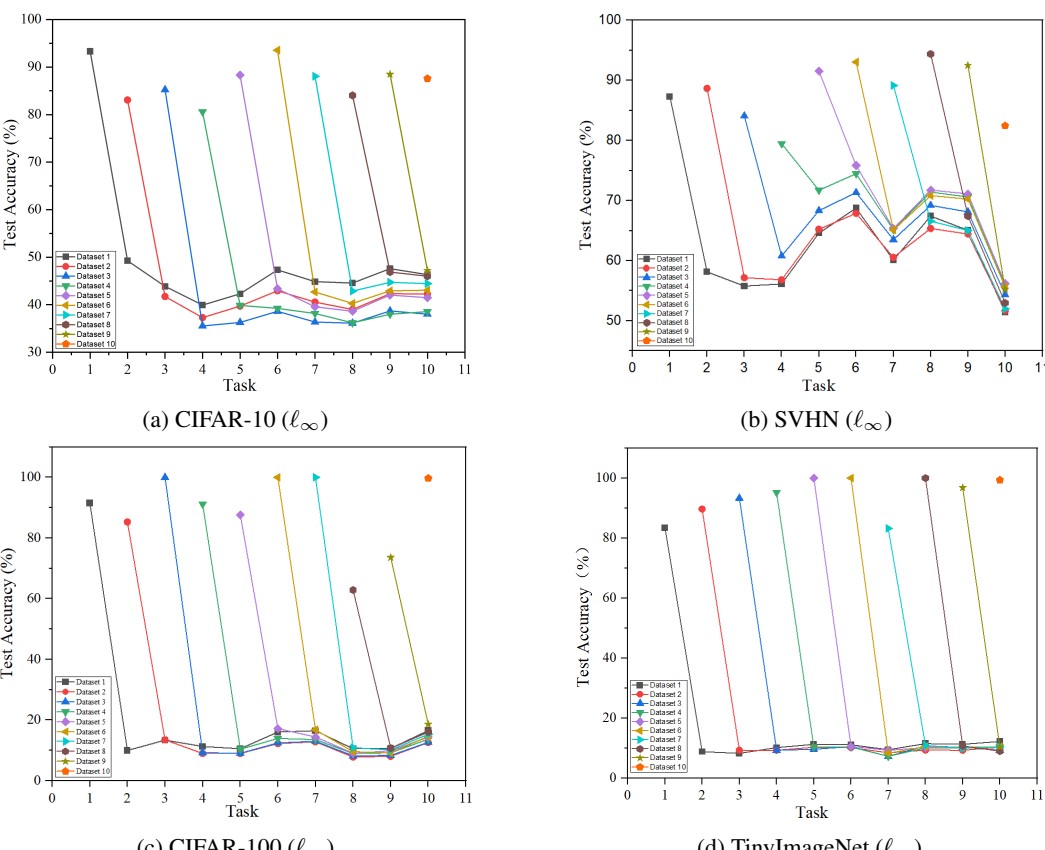

Figure 3: Catastrophic forgetting verification in the AT scenario on datasets CIFAR-10 ($\ell_\infty$), SVHN ($\ell_\infty$), CIFAR-100 ($\ell_\infty$), TinyImageNet ($\ell_\infty$). The horizontal axis represents a timestamp, where each number indicates the order of the current task.

From Figure 3, it is obvious that the phenomenon of forgetting is very clear, with the classifier ability on $\mathcal{D}_{<t}^{adv}$ sharply decreasing after fine-tuning on $\mathcal{D}_t^{adv}$. For instance, the test accuracy of $\mathcal{D}_1^{adv}$ dropped from 93.30% to 42.24% at the end of training on CIFAR-10. We consistently find that forgetting occurs across a variety of datasets (i.e., SVHN, CIFAR-10, CIFAR-100, and TinyImageNet), indicating that it is a general property of the adversarial training.

## B.2 CONNECTIONS BETWEEN AT AND CL

To visually demonstrate the dynamic nature of adversarial sample distributions during training, we employed t-SNE analysis (Van der Maaten & Hinton, 2008), a technique that effectively captures the complexity of high-dimensional data in a reduced-dimensional space. Specifically, we trained a PGD-10 model on the MNIST dataset using the PreActResNet-18 architecture under the $\ell_\infty$ norm with $\epsilon = 8/255$. We then performed t-SNE visualizations on test adversarial samples at 20-epoch intervals, from the beginning to the end of training. The results, shown in Figure 4, reveal distinct shifts in the data distribution of adversarial samples across epochs. This progression is similar to the challenges faced in continual learning, where models must adapt to evolving data distributions without forgetting previous states. This analysis visually substantiates the similarities between AT and continual learning processes.

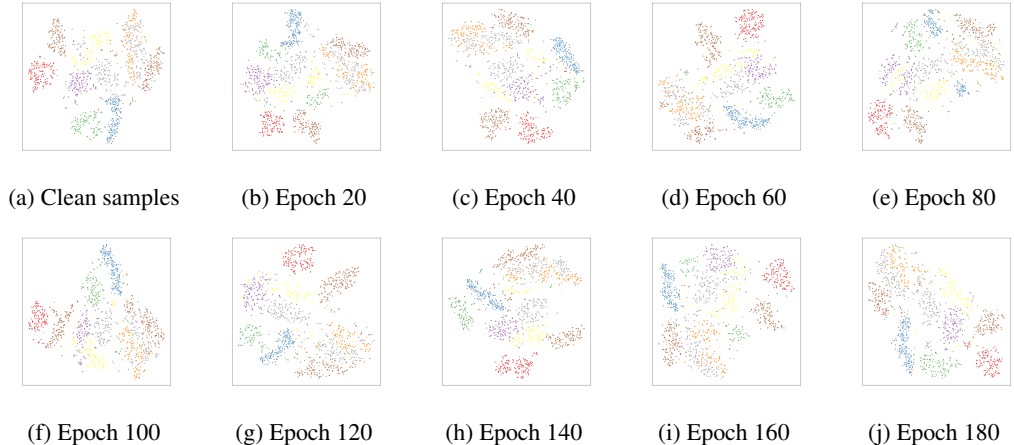

| (a) Clean samples | (b) Epoch 20 | (c) Epoch 40 | (d) Epoch 60 | (e) Epoch 80 |
| (f) Epoch 100 | (g) Epoch 120 | (h) Epoch 140 | (i) Epoch 160 | (j) Epoch 180 |

Figure 4: The t-SNE visualization of sample distributions, showing clean samples in (a) and adversarial samples from epoch 20 (b) to epoch 180 (j). Each plot illustrates the dynamic changes in the representation of adversarial samples as training progresses. we trained a PGD-10 model on the MNIST dataset under the $\ell_\infty$ norm with $\epsilon = 8/255$, using the PreActResNet-18 architecture.

## B.3 ADDITIONAL EXPERIMENT RESULTS FOR SENSITIVITY OF HYPER-PARAMETER

In Table 4, we provide the test accuracy of our method (AMS) with different regularization parameter $\lambda = 0.05, 0.1, 0.2, 0.3, 0.5, 1.0, 2.0, 3.0$ and interval parameter $m = 10, 20, 30, 50$ on CIFAR-10 with $\ell_\infty$ and $\epsilon = 8/255$ threat model using PreActResNet-18. In Table 5, we present the robust accuracy, clean accuracy, memory usage, and computation time for various values of $m$. Additionally, the corresponding curves are shown in Figure 5.

Table 4: Test accuracy (%) of the proposed method with different regularization parameter $\lambda = 0.05, 0.1, 0.2, 0.3, 0.5, 1.0, 2.0, 3.0$ and interval parameter $m = 10, 20, 30, 50$ on CIFAR-10 with $\ell_\infty$ and $\epsilon = 8/255$ threat model for PreActResNet-18. We highlight the best results in **bold**.

| $m$ | $\lambda$ | Clean | PGD-20 | CW-20 | AA | $\lambda$ | Clean | PGD-20 | CW-20 | AA |
|---|---|---|---|---|---|---|---|---|---|---|
| | 0.05 | 83.45 | 55.32 | 52.30 | 50.31 | 0.5 | 83.78 | **55.67** | **53.43** | **51.32** |
| 10 | 0.1 | 83.76 | 55.21 | 52.77 | 51.06 | 1.0 | 83.63 | 55.41 | 53.31 | 49.82 |
| | 0.2 | **84.25** | 55.36 | 52.64 | 51.03 | 2.0 | 83.34 | 55.21 | 53.14 | 49.79 |
| | 0.3 | 83.91 | 55.38 | 52.76 | 50.96 | 3.0 | 83.20 | 55.01 | 52.99 | 49.63 |
| | 0.05 | 83.98 | 55.13 | 52.23 | 50.30 | 0.5 | 84.86 | **55.59** | **53.31** | **51.22** |
| 20 | 0.1 | 84.12 | 55.20 | 52.70 | 51.03 | 1.0 | 84.45 | 55.06 | 53.24 | 49.78 |
| | 0.2 | 84.50 | 55.28 | 52.53 | 50.87 | 2.0 | 84.67 | 55.11 | 53.08 | 50.17 |
| | 0.3 | **84.93** | 55.41 | 52.78 | 50.95 | 3.0 | 84.21 | 54.78 | 52.89 | 49.94 |
| | 0.05 | 84.03 | 55.01 | 52.15 | 50.27 | 0.5 | 84.91 | **55.36** | **53.01** | **50.97** |
| 30 | 0.1 | 84.15 | 55.18 | 52.63 | 50.89 | 1.0 | 84.20 | 54.82 | 52.91 | 49.33 |
| | 0.2 | 84.58 | 55.19 | 52.66 | 50.95 | 2.0 | 84.36 | 54.93 | 52.87 | 49.36 |
| | 0.3 | **85.01** | 55.28 | 52.50 | 50.63 | 3.0 | 84.30 | 54.47 | 52.35 | 49.06 |
| | 0.05 | 83.21 | 54.88 | 52.07 | 50.16 | 0.5 | **85.12** | **55.34** | **53.16** | **51.04** |
| 50 | 0.1 | 84.13 | 55.08 | 52.53 | 50.67 | 1.0 | 84.58 | 55.03 | 53.00 | 49.52 |
| | 0.2 | 84.38 | 54.96 | 52.34 | 50.59 | 2.0 | 84.35 | 54.82 | 52.79 | 49.25 |
| | 0.3 | 84.97 | 55.04 | 52.58 | 50.49 | 3.0 | 84.30 | 54.63 | 52.51 | 49.57 |

Table 5: Robust and clean accuracy, memory usage, and computation time for different values of $m$.

| $m$ | AA (%) | Clean (%) | Memory (MB) | Time (h) |
|---|---|---|---|---|
| 10 | 51.32 | 83.78 | 853.6 | 23.12 |
| 20 | 51.22 | 84.86 | 426.8 | 17.56 |
| 30 | 50.97 | 84.91 | 256.08 | 15.77 |
| 50 | 51.04 | 85.12 | 170.72 | 13.01 |

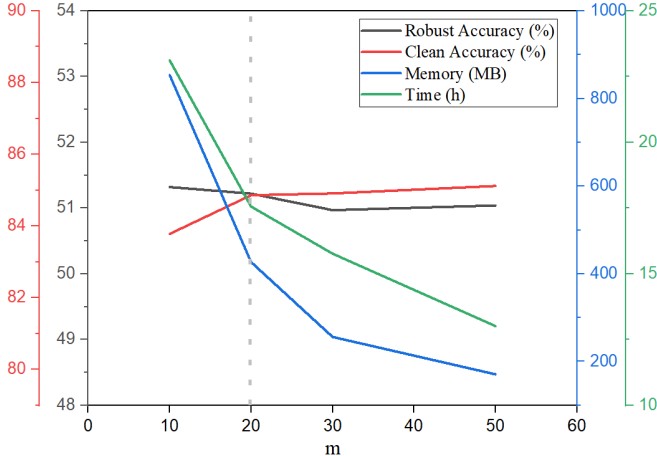

Figure 5: Evaluation of interval parameter from: memory usage, time consumption, clean accuracy, and robust accuracy.

## B.4 Additional Experiment Results for Robust Overfitting

**Comparison with Related Works.** In Table 7, we compare the performance of our method in terms of robust overfitting with a number of related works, such as AWP (Wu et al., 2020), SAT (Huang et al., 2020),TE (Dong et al., 2022), and LS (Pang et al., 2021). To empirically compare with these methods, we conduct experiments on CIFAR-10 with the PreActResNet-18 network under $\ell_\infty$ threat model. The results are presented in Table 7 from both the best checkpoint, which achieves the highest robust test accuracy under PGD-20 (denoted as "Best"), and the final checkpoint (denoted as "Final"), as well as the differential between these two checkpoints (denoted as "Diff"). A larger value of "Diff" indicates a more pronounced occurrence of robust overfitting. We observe that our AMS method has the potential to alleviate robust overfitting by addressing forgetting.

Table 6: Test accuracy (%) of the proposed methods and two baseline methods with regularization parameter $\lambda = 0.5$ and interval parameter $m = 20$ on CIFAR-10/100 and SVHN with $\ell_\infty$ and $\epsilon = 8/255$ threat model for PreActResNet-18. We highlight the best results in **bold**.

| Dataset | Method | Clean | | | PGD-20 | | | CW-20 | | | AA | | |
|---|---|---|---|---|---|---|---|---|---|---|---|---|---|
| | | Best | Final | Diff | Best | Final | Diff | Best | Final | Diff | Best | Final | Diff |
| CIFAR-10 | TRADES | 83.25 | 84.36 | -1.11 | 53.29 | 50.01 | 3.28 | 49.98 | 48.03 | 1.95 | 49.11 | 46.37 | 2.74 |
| | TRADES+AMS | **84.86** | **85.27** | **-0.41** | **55.59** | **55.01** | 0.58 | **53.31** | **52.86** | 0.45 | **51.22** | **50.57** | 0.65 |
| CIFAR-100 | TRADES | 58.26 | 59.83 | -1.57 | 31.07 | 28.84 | 2.23 | 27.85 | 26.02 | 1.83 | 25.93 | 24.36 | 1.57 |
| | TRADES+AMS | **60.03** | **60.67** | **-0.64** | **33.21** | **32.84** | 0.37 | **29.47** | **28.96** | 0.51 | **28.05** | **27.31** | 0.74 |
| SVHN | TRADES | **91.02** | **91.56** | -0.54 | 60.23 | 58.31 | 1.92 | 53.33 | 50.11 | 3.22 | 41.56 | 39.07 | 2.49 |
| | TRADES+AMS | 90.67 | 90.82 | **-0.15** | **60.45** | **59.84** | 0.61 | **56.07** | **55.67** | 0.40 | **52.64** | **51.77** | 0.87 |

Table 7: Test accuracy (%) of our proposed method and related works on the CIFAR-10 dataset using a PreActResNet-18 model within an $\ell_\infty$ threat model and $\epsilon = 8/255$. This experiment was conducted under the TRADES framework. We chose the best checkpoint based on the highest robust accuracy achieved on the test set under PGD-20. The best results are highlighted in **bold**.

| Method | Clean | | | PGD-20 | | | CW-20 | | | AA | | |
|---|---|---|---|---|---|---|---|---|---|---|---|---|
| | Best | Final | Diff | Best | Final | Diff | Best | Final | Diff | Best | Final | Diff |
| TRADES (Zhang et al., 2019b) | 83.25 | 84.36 | -1.11 | 53.29 | 50.01 | 3.28 | 49.98 | 48.03 | 1.95 | 49.11 | 46.37 | 2.74 |
| AWP (Wu et al., 2020) | 83.27 | 84.88 | -0.61 | 54.33 | 53.02 | 1.31 | 51.74 | 50.12 | 1.62 | 50.39 | 49.27 | 1.12 |
| SAT (Huang et al., 2020) | 82.97 | 82.36 | 0.61 | 53.24 | 52.86 | **0.38** | 52.35 | 51.88 | 0.47 | 50.46 | 49.87 | **0.59** |
| TRADES+TE (Dong et al., 2022) | 82.65 | 83.21 | -0.56 | 55.07 | 54.11 | 0.96 | 52.64 | 52.16 | 0.48 | 50.83 | 49.82 | 1.01 |
| TRADES+LS (Pang et al., 2021) | 83.05 | 84.14 | -1.09 | 53.86 | 51.65 | 2.21 | 51.35 | 50.24 | 1.11 | 49.76 | 47.82 | 1.94 |
| TRADES+AMS | **84.86** | **85.27** | -0.41 | **55.59** | **55.01** | 0.58 | **53.31** | **52.86** | 0.45 | **51.22** | **50.57** | 0.65 |

## B.5 Additional Experiment Details and Results for CL

**Details for experimental setup.** Most of the comparison methods (CL) in our experiments use the implementation of Mammoth[1]. For replay-based method such as ER (Chaudhry et al., 2019), DER (Buzzega et al., 2020), GCR (Tiwari et al., 2022) and DGC-ER (Lin et al., 2024), we set the memory size to 500,000, which allows us to store 500,000 samples. For regularization-based methods such as EWC (Kirkpatrick et al., 2017) and Lwf (Li & Hoiem, 2017), we set the interval parameter $m$ to 20, the same as our method. In Table 8 and Table 9, we present the test accuracy of our proposed method and other CL methods on CIFAR-10 with an $\ell_\infty$ and $\epsilon = 8/255$ threat model using PreActResNet-18 within the PGD and TRADES frameworks.

---

[1] https://github.com/aimagelab/mammoth

Table 8: Test accuracy (%) of the proposed methods and other continual learning methods on CIFAR-10 with an $\ell_\infty$ and $\epsilon = 8/255$ threat model for PreActResNet-18 under the PGD-10 framework. The best results are highlighted in **bold**.

| Method | Clean | PGD-20 | CW-20 | AA |
|---|---|---|---|---|
| PGD-10 | **85.29** | 52.31 | 51.67 | 47.70 |
| PGD-10+ER (Chaudhry et al., 2019) | 82.72 | 52.44 | 50.85 | 47.81 |
| PGD-10+DER (Buzzega et al., 2020) | 82.78 | 52.51 | 50.93 | 48.02 |
| PGD-10+GCR (Tiwari et al., 2022) | 82.21 | 51.39 | 49.85 | 47.11 |
| PGD-10+EWC (Kirkpatrick et al., 2017) | 83.21 | 53.87 | 51.24 | 49.53 |
| PGD-10+Lwf (Li & Hoiem, 2017) | 83.84 | 54.76 | 52.82 | 50.45 |
| PGD-10+DGC-ER (Lin et al., 2024) | 82.95 | 52.58 | 51.10 | 48.11 |
| PGD-10+**AMS** | 84.45 | **55.83** | **53.17** | **51.63** |

Table 9: Test accuracy (%) of the proposed methods and other continual learning methods on CIFAR-10 with an $\ell_\infty$ and $\epsilon = 8/255$ threat model for PreActResNet-18 under the TRADES framework. The best results are highlighted in **bold**.

| Method | Clean | PGD-20 | CW-20 | AA |
|---|---|---|---|---|
| TRAEDS | 83.25 | 53.29 | 49.98 | 49.11 |
| TRAEDS+ER (Chaudhry et al., 2019) | 80.86 | 53.47 | 50.33 | 49.82 |
| TRAEDS+DER (Buzzega et al., 2020) | 81.01 | 53.55 | 50.47 | 50.05 |
| TRAEDS+GCR (Tiwari et al., 2022) | 80.92 | 53.31 | 49.84 | 48.64 |
| TRAEDS+EWC (Kirkpatrick et al., 2017) | 83.75 | 54.15 | 51.73 | 50.37 |
| TRAEDS+Lwf (Li & Hoiem, 2017) | 84.32 | 54.99 | 52.86 | 50.68 |
| TRAEDS+DGC-ER (Lin et al., 2024) | 81.17 | 53.69 | 50.53 | 50.12 |
| TRAEDS+**AMS** | **84.86** | **55.59** | **53.31** | **51.22** |

## B.6 OTHER EMPIRICAL DETAILS AND RESULTS

**Adversarial training with $\ell_2$ threat model.** In Table 10, we present the test accuracy of the proposed method and PGD-10 on the CIFAR-10, CIFAR-100, and SVHN datasets under an $\ell_2$ threat model with $\epsilon = 128/255$, using the PreActResNet-18 architecture within the PGD framework.

**Robustness under black-box attacks.** We train a PreActResNet-18 model under the $\ell_\infty$ threat model on the CIFAR-10, CIFAR-100, and SVHN datasets. Black-box adversarial examples are generated using a surrogate adversarial training model with identical settings, employing PGD-20 and CW-20 attacks. As shown in Table 11, the results demonstrate that our method can effective defense black-box attacks.

**Alleviate forgetting.** In Table 12, we validate the effectiveness of our method in mitigating forgetting. Compared to Table 1, it is evident that our method significantly alleviates forgetting.

**Different structures.** In Table 13, we apply AMS to other structures, including VGG16 (Simonyan, 2014) and ViT (Dosovitskiy et al., 2021). The results illustrate that our AMS method is effective on these structures as well.

Table 10: Test accuracy (%) of the proposed methods and PGD-10 on CIFAR-10/100 and SVHN with an $\ell_2$ and $\epsilon = 128/255$ threat model for PreActResNet-18. The results of our methods are in **bold**.

| Dataset | Method | Clean | PGD-20 |
|---|---|---|---|
| CIFAR-10 | PGD-10 (Madry et al., 2018) | 87.85 | 70.28 |
|  | PGD-10+AMS | **88.21** | **72.95** |
| CIFAR-100 | PGD-10 (Madry et al., 2018) | 60.05 | 43.52 |
|  | PGD-10+AMS | **64.88** | **45.02** |
| SVHN | PGD-10 (Madry et al., 2018) | 92.95 | 72.85 |
|  | PGD-10+AMS | **93.25** | **73.10** |

Table 11: Black-box test accuracy (%) of the proposed methods and PGD-10 on CIFAR-10/100 and SVHN with an $\ell_\infty$ and $\epsilon = 8/255$ threat model for PreActResNet-18. The results of our methods are in **bold**.

| Method | CIFAR-10 | | | CIFAR-100 | | | SVHN | | |
|---|---|---|---|---|---|---|---|---|---|
|  | Clean | PGD | CW | Clean | PGD | CW | Clean | PGD | CW |
| PGD-10 (Madry et al., 2018) | 85.29 | 58.52 | 56.31 | 58.12 | 36.15 | 33.07 | 89.12 | 58.92 | 54.46 |
| PGD-10+AMS | **85.45** | **62.38** | **60.47** | **58.87** | **39.73** | **37.56** | **90.77** | **66.23** | **63.64** |

Table 12: Verification results of the forgetting phenomenon in adversarial training across different datasets and perturbation threat models, integrating with our AMS approach. It is important to note that the adversarial datasets accumulated at each phase are initially classified with perfect accuracy, thereby presenting an initial accuracy rate of 100% for each dataset.

| Dataset | Norm | Robust Test Accuracy (%) | | | | | FAA (%) | FF (%) |
|---|---|---|---|---|---|---|---|---|
|  |  | Stage 1 | Stage 2 | Stage 3 | Stage 4 | Stage 5 |  |  |
| CIFAR-10 | $\ell_\infty$ | 92.54 | 94.28 | 95.77 | 97.82 | 98.20 | 95.72 | 4.28 |
|  | $\ell_2$ | 90.83 | 96.71 | 97.84 | 95.82 | 96.25 | 95.49 | 4.51 |
| CIFAR-100 | $\ell_\infty$ | 86.17 | 96.59 | 96.35 | 93.13 | 95.17 | 93.48 | 6.52 |
|  | $\ell_2$ | 87.75 | 95.73 | 95.34 | 96.40 | 97.65 | 94.57 | 5.43 |
| SVHN | $\ell_\infty$ | 92.17 | 97.36 | 97.67 | 98.20 | 97.89 | 96.66 | 3.34 |
|  | $\ell_2$ | 94.62 | 94.42 | 97.64 | 97.74 | 98.14 | 96.51 | 3.49 |
| TinyImageNet | $\ell_\infty$ | 72.53 | 86.81 | 83.18 | 87.59 | 88.56 | 83.73 | 16.27 |
|  | $\ell_2$ | 75.37 | 86.90 | 85.63 | 88.93 | 92.23 | 85.81 | 14.19 |

Table 13: Test accuracy (%) on CIFAR-10 with $\ell_\infty$ and $\epsilon = 8/255$ threat model for ViT and VGG16.

| Architecture | Method | Clean | AA |
|---|---|---|---|
| ViT | TRADES (Zhang et al., 2019b) | 85.11 | 47.04 |
|  | TRADES+AMS | 85.36 | 49.62 |
| VGG16 | TRADES (Zhang et al., 2019b) | 79.66 | 44.19 |
|  | TRADES+AMS | 81.03 | 45.79 |

