# OpenReview forum: "Exploring The Forgetting in Adversarial Training: A Novel Method for Enhancing Robustness"
_ICLR.cc/2025/Conference — ICLR 2025 Poster_

### Official Review · Reviewer_rCAv · 2024-10-18

**Soundness:** 3
**Presentation:** 2
**Contribution:** 3
**Rating:** 6
**Confidence:** 3

**Summary:**

This work proposes an enhanced adversarial training algorithm that tackles this problem from the perspective of continual learning and catastrophic forgetting. It first demonstrates a forgetting phenomenon with an experiment. Then, a methodology named AMS that distills from weighted teachers (checkpoints from earlier training epochs) is proposed. Empirical experiments on SVHN, CIFAR-10/100, and Tiny ImageNet show improved robustness when adding AMS upon e.g. Madry's adversarial training and TRADES. Robust overfitting can also be mitigated with the help of AMS.

**Strengths:**

The work is presented in a clear and smooth way. Viewing adversarial training from a continual learning and forgetting perspective is interesting and could be of significance if it's novel (which I'm not entirely sure since I'm away from adversarial training literature for a while). It starts by exposing the forgetting problem with a concrete experiment, which justifies the validity of employing the forgetting view to some extent. The method is also presented well, with Figure 2 being quite informative. Experiments seem quite extensive.

**Weaknesses:**

1. AMS will inevitably incur larger time and space cost. Beyond increasing $m$, is there any other more intelligent method that can help reduce the number of teachers? For example, could there be a metric that helps identify the most informative teachers, so that throughout the training one can only keep say the most valuable $k$ teachers?

2. One straightforward approach for aggregating or maintaining learned information from previous models is Exponential Moving Average (EMA), which has been shown to help with adversarial training [1]. EMA should be considered as a baseline in my opinion.

3. An ablation study should be presented to validate the proposed reweighting-based loss correction.

4. In Table 6, it seems that distilling from more teachers (having smaller $m$) hurts clean accuracy but not robustness. Any idea why?


[1] Uncovering the Limits of Adversarial Training against Norm-Bounded Adversarial Examples

**Questions:**

Please see the questions in weaknesses.

---

> ### Author Response · Authors · 2024-11-22
>
> Thank you for your thoughtful comments and suggestions, and we try to address your questions and concerns below.
>
> **W1:  AMS will inevitably in ...  valuable k teachers?**
>
> **A:** Thank you for this insightful suggestion. Yes, we think developing a metric to identify the most informative teachers is a promising approach to reduce the number of teachers retained. Identifying the most valuable teachers is indeed a complex task, as the impact of a teacher can vary significantly across different stages of student training. Several factors should be considered, including the relevance of the teacher's knowledge to current training needs and the diversity of the knowledge they offer. In our future work, we plan to design such a metric into the AMS framework. This would enable us to dynamically select a limited number of k most valuable teachers at any given point, based on their assessed impact and relevance.
>
> **W2: One straightforward ... EMA should be considered as a baseline in my opinion.**
>
> **A:** Thanks for your suggestion. We conducted the following new experiment. We incorporated EMA into our experimental setup, implementing the update rule $\theta' \leftarrow \gamma \cdot \theta' + (1-\gamma)\cdot \theta$ at each training step with a decay rate $\gamma = 0.995$. During the evaluation, we used the weighted parameters $\theta'$ instead of the directly trained parameters $\theta$. The results of this inclusion are presented in Table 1 below, where we compare the performance of TRADES enhanced with EMA against TRADES enhanced with our AMS method. As shown, AMS consistently outperforms EMA across both clean and robust accuracy metrics on CIFAR-10 and CIFAR-100 datasets.
>
> Table 1: Test accuracy (%) of the proposed methods and TRADES + EMA methods on CIFAR-10/100 under the $\ell_\infty$ norm with $\epsilon = 8/255 $ based on the WideResNet-34-10 architecture.
>
> | Method       | CIFAR-10 (clean) | CIFAR-10 (AA) | CIFAR-100 (clean) | CIFAR-100 (AA) |
> | ------------ | ---------------- | ------------- | ----------------- | -------------- |
> | TRADES + EMA | 84.78            | 53.76         | 61.43             | 27.19          |
> | TRADES + AMS | 85.37            | 54.31         | 62.56             | 28.25          |

---

> ### Author Response · Authors · 2024-11-22
>
> **W3: An ablation study should be presented to validate the proposed reweighting-based loss correction.**
>
> **A:** Thanks for raising this question. Following your suggestion, we have conducted the ablation experiments and included the results in the Table 2 below. These experiments specifically assess the impact of the RLC component on the robust and clean accuracies of models trained with TRADES+AMS and PGD-10+AMS. The results demonstrate that the inclusion of the RLC can enhance the robust accuracy across both CIFAR-10 and CIFAR-100 datasets datasets to certain extent. For example, when RLC is removed, the robust accuracy on CIFAR-100 decreases from 28.05% to 27.62% compared with TRADES+AMS, and from 28.31% to 27.58% compared with PGD-10+AMS, illustrating the substantial contribution of the RLC component to enhancing adversarial robustness.
>
> Table 1: Test accuracy (%) of TRADES+AMS and PGD-10+AMS with or without (w/o) the ''Reweighting-based Loss Correction'' (RLC) term under the $\ell_{\infty}$ norm with $\epsilon=8/255$ based on the PreActResNet-18 architecture.
>
> | Method             | CIFAR-10 (clean) | CIFAR-10 (AA) | CIFAR-100 (clean) | CIFAR-100 (AA) |
> | ------------------ | ---------------- | ------------- | ----------------- | -------------- |
> | TRADES+AMS         | 84.86            | 51.22         | 60.03             | 28.05          |
> | TRADES+AMS w/o RLC | 84.72            | 50.08         | 60.11             | 27.62          |
> | PGD-10+AMS         | 85.45            | 51.63         | 58.87             | 28.31          |
> | PGD-10+AMS w/o RLC | 85.66            | 50.31         | 58.83             | 27.58          |
>
> **W4:  In Table 6, it seems that distilling from more teachers (having smaller m) hurts clean accuracy but not robustness. Any idea why?**
>
> **A:** Thank you for this insightful question. In our method, the distillation loss $\mathcal{L}_{ams}$ (Eq. 8 in page 7) is calculated using the adversarial samples $\mathbf{x}_i'$ to preserve the knowledge from previous stages and enhance the model’s robustness. This approach is effective in preventing catastrophic forgetting and enhancing robustness, but it also introduces a minor problem: as the number of teachers increases, the student model is exposed to a broader range of potentially conflicting information during the distillation process. Each teacher model may emphasize different aspects of the input data, which could lead to a less coherent overall signal for the student model to follow. This diversity in guidance can complicate the optimization landscape, particularly when trying to generalize from clean data, thereby impacting clean accuracy. Therefore, we consider both clean accuracy and robust accuracy when determining the hyperparameter $m$ in our experiment, as shown in Figure 4 on page 21.

---

> ### Comment · Reviewer_rCAv · 2024-11-22
> **Thank you for the response**
>
> The rebuttal sufficiently addressed my comments. I don't have other questions.

---

> > ### Author Response · Authors · 2024-11-23
> >
> > We  thank the reviewer again for the helpful comments.

---

### Official Review · Reviewer_XV9q · 2024-10-29

**Soundness:** 3
**Presentation:** 3
**Contribution:** 3
**Rating:** 8
**Confidence:** 5

**Summary:**

This paper studies the adversarial learning problem from a continual learning perspective. In particular, the authors found a novel connection between catastrophic forgetting; a well-studied problem in continual learning, and adversarial training (where the model at later stages forgets the learnt adversaries from earlier stages). Based on this connection, the authors propose a novel method that distills the knowledge from multiple teachers during adversarial training. Each teacher is a previously saved checkpoint obtained through the adversarial training. Through extensive experiments on different datasets and threat models, the proposed orthogonal method shows consistent performance improvement.

**Strengths:**

This paper has the following strengths:

1- The connection between adversarial training and continual learning is the part that I like the most in this paper. This should open the door for further research on the intersection between these two fields.

2- The method proposed in this paper is intuitive, and more importantly is orthogonal. That is, if a stronger training algorithm is developed in the future, one would expect that this algorithm + AMS would perform better.

3- The experiments are comprehensive spanning multiple datasets, adversarial training schemes, and threat models.

**Weaknesses:**

Despite the strengths in this paper, there are still remaining concerns that need to be resolved.

1- Concern regarding experiments: In Table 2, it seems that S$^2$O performs better than TRADES+AMS in every aspect on CIFAR-10 (both clean accuracy and robust accuracy under all considered attacks). This then begs for the following two experiments:

1a. Can yo provide the performance of S$^2$O on the other considered datasets (e.g. CIFAR100, SVHN, TinyImageNet)?

1b. Can you provide the results of S$^2$O + AMS? Generally, how is the performance of other robust training methods when combined with AMS?

There are also a couple of other experiments missing in this paper; namely:

1c. Experiments on CIFAR10 using the 1M generated images (with and without AMS).

1.d Comparison agains AWP [A] seems to be missing in this paper.

2- Concern regarding presenting results in table: I think it is quite confusing to not make the best results in bold rather than making the proposed method's results in bold. I think an easier way to trace the results is to distinguish the best results and the runner up. Alternatively, you could present each method as (Method in one row, Method + AMS in another row highlighting the performance gain under AMS).

3- Concern about the proposed method: The results in the appendix (Table 5) show that AMS can suffer from significant performance drop under different choices of hyper parameters (namely $\lambda$). Is the choice of hyper parameters at least generalizable under different robust training methods? A simple experiment to address this concern is to fix the hyper parameters for the suggested experiment in 1.b to be $m=20, \lambda=0.5$.

Minor comments:

4- Performance gain under AMS is marginal (often ~1%).

5- How is the performance of AMS in the continual learning setting? I am curios to see if it actually outperforms the CL methods such as LwF and ER.

6- Missing reference: I find the motivational experiments of this paper (experiments on adversarial examples from previous checkpoints) are quite similar to the work of Gupta et.al [B]. I believe that the related work in this paper will benefit from a discussion on the relation between the results in Figure 1 in this work and the results in Figure 1 in [B].

[A] Adversarial Weight Perturbation Helps Robust Generalization, NeurIPS 2020

[B] Improving the affordability of robustness training for DNNs, CVPRW 2020

**Questions:**

Please refer to the weaknesses section. I am happy to raise my score if my questions and concerns are resolved.

---

> ### Author Response · Authors · 2024-11-22
>
> Thank you for your thoughtful comments and suggestions, and we try to address your questions and concerns below.
>
> **W1: Concern regarding experiments: In Table 2, it seems that $S^2O$ performs better than TRADES+AMS in every aspect on CIFAR-10 (both clean accuracy and robust accuracy under all considered attacks). This then begs for the following two experiments: 1a and 1b. There are also a couple of other experiments missing in this paper; namely 1c and 1d.**
>
> **A:** Thanks for your questions regarding several key aspects of our experimental validation.
>
> **1a. Performance of $S^2O$ on Other Datasets:** Responding to your inquiry about the performance of $S^2O$ on other datasets, we have extended our experiments to include CIFAR-100, SVHN, and TinyImageNet, alongside CIFAR-10. The results are shown in Table 1 below.
>
> **1b. Integration of AMS with $S^2O$ and Other Robust Training Methods:** Furthermore, to address your second point, we explored the integration of AMS with $S^2O$. The results, detailed in Table 1 below, show that combining AMS with $S^2O$ consistently improves robust accuracy on all tested datasets. This observation aligns with our findings that AMS can enhance the robustness of other training methodologies. Typically, we observe an approximate increase in robust accuracy by ~2% when AMS is applied to various robust training methods.
>
> Table 1: Test accuracy (%) of $S^2O$ and $S^2O$ + AMS under the $\ell_{\infty}$ norm with $\epsilon=8/255$ based on the **WideResNet-34-10**  architecture.
>
> | Method      | CIFAR-10 (clean) | CIFAR-10 (AA) | CIFAR-100 (clean) | CIFAR-100 (AA) | TinyImageNet (clean) | TinyImageNet (AA) | SVHN (clean) | SVHN (AA) |
> | ----------- | ---------------- | ------------- | ----------------- | -------------- | -------------------- | ----------------- | ------------ | --------- |
> | $S^2O$      | 85.58            | 53.93         | 63.31             | 27.50          | 52.11                | 21.63             | 93.56        | 58.12     |
> | $S^2O$ +AMS | 85.83            | 55.92         | 63.67             | 29.96          | 52.40                | 22.85             | 93.37        | 60.62     |
>
> **1c. Experiments with 1M Generated Images on CIFAR-10:** In response to your query, we have conducted additional experiments using 1M generated images on CIFAR-10, both with and without the AMS method. The results are presented in Table 2 below, which shows that incorporating AMS can enhance robust accuracy and meanwhile preserve high clean accuracy.
>
> **1d. Comparison Against AWP:** Regarding the comparison with the Adversarial Weight Perturbation (AWP) method, while we discussed AWP in Tables 7 and 8 within Section 5.3 (about Robust Overfitting), we further conducted more experiments to directly compare it against our AMS method, and show the results in the Table 2. The addition of AMS to AWP can improve the robust accuracy , highlighting the advantage of AMS when integrated with other robust training methods.
>
> Table 2: Test accuracy (%) of different methods under the $\ell_{\infty}$ norm with $\epsilon=8/255$  on CIFAR-10, using the **WideResNet-34-10**  architecture.
>
> | Method                           | clean | AA    |
> | -------------------------------- | ----- | ----- |
> | Wang et al. (2023) (1M) [1]      | 91.18 | 63.31 |
> | Wang et al. (2023) (1M) [1] +AMS | 90.79 | 64.97 |
> | AWP                              | 83.45 | 50.83 |
> | AWP +AMS                         | 84.66 | 53.17 |

---

> ### Author Response · Authors · 2024-11-22
>
> **W2: Concern regarding presenting results in table:...performance gain under AMS).**
>
> **A:** We appreciate your feedback on the table formatting and we will make these modifications in the final version.
>
> **W3: Concern about the proposed method: ... in 1.b to be $m=20$, $\lambda=0.5$.**
>
> **A:** Yes, regarding your concern about this question, we fixed the hyperparameters for the experiment in **W1 1b**. It can be found that the choice of hyperparameters is generalizable under different robust training methods.
>
> **W4: Performance gain under AMS is marginal (often ~1%).**
>
> **A:** Thank you for the comment. We acknowledge that the numerical improvements may appear modest in some cases, but AMS also demonstrates significant enhancements in a number of other cases, highlighting its advantage under common adversarial training methods. For example, it achieves a ~4% improvement in robust accuracy for PGD-10 on CIFAR-10, as shown in Table 3 of our manuscript. Moreover, another highlight of our method is that it proposes a new perspective for enhancing robustness during AT, and we hope this idea can inspire more improvement along this line in future.
>
> **W5: How is the performance of AMS in the continual learning setting? ... such as LwF and ER.**
>
> **A:** We think it is an interesting question that how AMS compares to established continual learning methods like Learning without Forgetting (LwF) and Experience Replay (ER). To address your query, we specifically designed a preliminary experiment using the Sequential-MNIST dataset, which splits the MNIST digits into five sequential tasks, introducing two new digits per task. The results of these experiments are summarized in Table 3. While AMS is not particularly designed for continual learning, it still achieves certain improvement upon LwF (though it is worse than ER). We believe the major advantage of our AMS method still lies in the enhancement for AT as shown in our manuscript.
>
> Table 3: Average accuracy at the end of all tasks on Sequential-MNIST dataset. The Sequential-MNIST protocol splits the entire training set of the MNIST Digits dataset into 5 tasks, with each task introducing two new digits.
>
> | Method | Average accuracy |
> | ------ | ---------------- |
> | Lwf    | 91.89            |
> | ER     | 97.55            |
> | AMS    | 93.11            |
>
> **W6: Missing reference: I ...  the results in Figure 1 in [B].**
>
> **A:** Thank you for bringing this to our attention, and we appreciate your pointing out the similarities between their motivational experiments and ours. We will thoroughly review the findings of Gupta et al. and provide a detailed discussion in the related work section of our manuscript. Specifically, we will compare the adversarial example generation methods and the results depicted in Figure 1 of both our work and that of Gupta et al.

---

> > ### Comment · Reviewer_XV9q · 2024-11-22
> > **Response to rebuttal**
> >
> > I would really like to thank the authors for their efforts in the rebuttal.
> >
> > I have the following two remaining concerns unresolved:
> >
> > (1) Regarding the experiments on AWP: I would like to thank the reviewer once again in conducting such experiments. However, it seems that the baseline numbers do not match the reported on the auto-attack repo (50.83 vs 56.17) despite the match in both the dataset and the architecture. Does the authors have any explanation on that regard?
> >
> > (2) It would be more convincing to reflect the suggested/requested edits (e.g. the discussion regarding the related reference) in the paper during the rebuttal period than promising it in the final version.

---

> > > ### Author Response · Authors · 2024-11-23
> > >
> > > **Q1:Regarding the experiments on AWP: ... Does the authors have any explanation on that regard?**
> > >
> > > **A:** Thanks for pointing out this issue. The discrepancy in the baseline numbers primarily arises from the different network architectures used in our experiments compared to those referenced. Specifically, we used ResNet-18, whereas the auto-attack repository results were based on WideResNet-34-10. We acknowledge that this was not accurately described in our original response. Therefore, we have updated Table 2 in response '1d' to include a column specifying the neural network architectures used in each case. Here is the revised Table 2:
> > >
> > > | Method                           | **Architecture** | clean | AA    |
> > > | :------------------------------- | ---------------- | :---- | :---- |
> > > | Wang et al. (2023) (1M) [1]      | WideResNet-34-10 | 91.18 | 63.31 |
> > > | Wang et al. (2023) (1M) [1] +AMS | WideResNet-34-10 | 90.79 | 64.97 |
> > > | AWP                              | ResNet-18        | 83.45 | 50.83 |
> > > | AWP +AMS                         | ResNet-18        | 84.66 | 53.17 |
> > >
> > > **Q2: It would be more convincing to reflect the suggested/requested edits (e.g. the discussion regarding the related reference) in the paper during the rebuttal period than promising it in the final version.**
> > >
> > > **A:** Thanks for your suggestion. We will upload a revised manuscript as soon as possible (within 1-2 days).

---

> ### Author Response · Authors · 2024-11-27
>
> Dear Reviewer XV9q,
>
> Thank you for taking the time to review our paper and provide valuable feedback. As the discussion phase is nearing its conclusion, we would like to confirm if our responses from a few days ago have effectively addressed your concerns. If you have any additional comments, we will do our best to address them.
>
> Best regards,
>
> The authors

---

> > ### Comment · Reviewer_XV9q · 2024-11-28
> > **Thank you**
> >
> > Dear Authors
> >
> > Thank you once again for engaging in the discussion and addressing my concern.
> >
> > Thus, I am raising my score.

---

> ### Author Response · Authors · 2024-11-29
>
> We thank the reviewer again for the positive and encouraging feedback.

---

### Official Review · Reviewer_VMqL · 2024-10-30

**Soundness:** 3
**Presentation:** 3
**Contribution:** 2
**Rating:** 6
**Confidence:** 4

**Summary:**

The paper presents a novel perspective on mitigating catastrophic forgetting during adversarial training. The paper proposes Adaptive Multi-teachers Self-distillation that uses an adaptive regularizer to align the knowledge differences across stages. The evaluation is performed on multiple target models under various gradient-based attacks.

**Strengths:**

1. The paper is well-written with a clear storyline and a sound motivation.
2. The proposed method is reasonable and fair.
3. The experiments on three attacks are comprehensive.

**Weaknesses:**

1. The proposed framework shallowly touches on the training efficiency of adversarial training. It is appreciated that authors mention effective variants of adversarial training that have high training efficiency (e.g., Free-AT [1], YOPO [2]). Yet the paper's algorithm design does not fully address how such efficiency techniques are involved in the framework. Also, Line 6 of Algorithm 1 adds an initial random perturbation to the source image, which is similar to [3]. Authors are expected to elaborate on the design choice of the $0.001$ strength used in this line.

2. The scope of target models is limited. The paper only evaluates CNN-based architectures (e.g., ResNet variants), while more modern vision models based on Vision Transformers are not sufficiently evaluated. It will help to validate the framework if experiments on ViT also support the existing findings.

[1] Adversarial Training for Free. NeurIPS 2019.

[2] You Only Propagate Once: Accelerating Adversarial Training via Maximal Principle. NeurIPS 2019.

[3] Fast is better than free: Revisiting adversarial training. ICLR 2020.

**Questions:**

Please address my concerns stated in the weakness section. Note that, although there are concerns raised, I believe none of them is severely affecting the validity of the paper. Hence I rate the initial version of the submission as a borderline accept. My final rating would be conditioned on the soundness of the authors' responses.

---

> ### Author Response · Authors · 2024-11-22
>
> Thank you for your thoughtful comments and suggestions, and we try to address your questions and concerns below.
>
> **W1: The proposed framework shallowly ... strength used in this line.**
>
> **A:** Thanks for your questions that help us to clarify our method.
>
> **Integration of Efficient Training Techniques:** Our primary contribution focuses on enhancing adversarial robustness by mitigating catastrophic forgetting. Nonetheless, integrating efficient training techniques is also deserved to study. Our AMS approach is designed as a plugin that can be flexibly used to enhance existing adversarial training methods. In our framework, the algorithms like 'Free adversarial training' [1] and 'YOPO' [2] can be effectively incorporated to optimize training efficiency. Specifically, by adjusting the interval parameter $m$ to sync with hop steps in 'Free-AT' and combining this with our AMS loss, we can enhance training speed while still alleviating forgetting and boosting robustness. This integration is described in the algorithmic pseudocode available in our anonymous supplementary material ([link to pseudocode](https://anonymous.4open.science/r/xxx-xxx/iclr.pdf)) (If garbled, please refresh). Given the time limit of rebuttal, we plan to add more complete study on these methods later.
>
> **Initial Random Perturbation in Algorithm 1:** Regarding the initial random perturbation added to the source image in Line 6 of Algorithm 1, this step is distinct from the technique mentioned in [3]. Our approach is essential for addressing the inner maximization problem of Eq.(3) through the projected gradient descent (PGD) method. In this context, $x_i$ represents a global minimizer with a **zero gradient** to the inner objective function. To effectively kick-start the inner optimizer, we initialize $x_{i}'$ by adding a slight, random perturbation around $x_i$, ensuring the PGD does not start at a stationary point.
>
> **W2: The scope of target models is limited. ...  framework if experiments on ViT also support the existing findings.**
>
> **A:** Thank you for highlighting this important aspect. We agree that the evaluation of our method using only CNN-based architectures, such as ResNet variants, limits the generalizability of our findings. To address this concern and enrich our study, we have conducted several extra experiments.
>
> We expanded our experimental setup to include Vision Transformer (ViT) and VGG16 architectures to assess the efficacy of our AMS under the $\ell_{\infty}$ norm with $\epsilon = 8/255$ on the CIFAR-10 dataset. The results of these experiments are presented in Table 1 below, showing both clean and robust accuracy (AA) improvements, which help validate the effectiveness of our framework across different model architectures. Given the time limit of rebuttal, we plan to add more complete study on these architectures later.
>
> Table 1: Test accuracy (%) on CIFAR-10 with $\ell_{\infty}$  and  $\epsilon=8/255$ threat model for ViT and VGG16.
>
> | Method              | clean | AA    |
> | ------------------- | ----- | ----- |
> | ViT (TRADES)        | 85.11 | 47.04 |
> | VIT (TRADES + AMS)  | 85.36 | 49.62 |
> | VGG16 (TRADES)      | 79.66 | 44.19 |
> | VGG16 (TRADES +AMS) | 81.03 | 45.79 |

---

> ### Author Response · Authors · 2024-11-27
>
> Dear Reviewer VMqL,
>
> Thank you for taking the time to review our paper and provide valuable feedback. As the discussion phase is nearing its conclusion, we would like to confirm if our responses from a few days ago have effectively addressed your concerns. If you have any additional comments, we will do our best to address them.
>
> Best regards,
>
> The authors

---

### Official Review · Reviewer_THQc · 2024-11-03

**Soundness:** 2
**Presentation:** 2
**Contribution:** 3
**Rating:** 6
**Confidence:** 4

**Summary:**

The authors approach AT from a continual learning perspective, treating adversarial samples generated with different parameters during training as tasks with distinct distributions, as models undergoing AT tend to forget adversarial examples learned in earlier stages, similar to the forgetting phenomenon in continual learning. This work introduces an Adaptive Multi-teacher Self-distillation method, which mitigates forgetting by aligning model outputs across training stages. Experiments on multiple datasets and under various adversarial attacks demonstrate that AMS enhances both clean and robust accuracy, while also alleviating robust overfitting.

**Strengths:**

1.	The approach of viewing AT as a continual learning problem is innovative, offering a new perspective on enhancing model robustness within AT.
2.	The paper is well-structured and clearly presents the progression from problem formulation to solution, supported by extensive experiments on diverse datasets that validate the proposed method’s robustness.

**Weaknesses:**

1.	The authors approach AT through the lens of continual learning but do not sufficiently establish the connection between the two, either experimentally or theoretically. For example, demonstrating how adversarial samples generated with different parameters follow distinct data distributions could strengthen the argument. Some statements appear subjective, despite the proposed method's effectiveness.
2.	The AMS method requires tracking knowledge across multiple stages, which increases memory and computational costs, particularly with large datasets and models.

**Questions:**

1.	Is the classification result shown in Figure 1 based on the training set? If so, this result may not fully illustrate the catastrophic forgetting problem; it may only indicate that newly generated samples were successfully attacked, especially given that an untargeted attack was used. If the result is on the test set, it would more convincingly suggest similarity to catastrophic forgetting in continual learning.
2.	Catastrophic overfitting[1] is a common issue in fast adversarial training, with many works focusing on its mitigation[2][3]. Have you explored whether the proposed method could address catastrophic overfitting as well, given its similarity to catastrophic forgetting in continual learning?
3.	Did the author consider dividing the epochs into stages unevenly, with each stage containing a different number of training epochs?
4.	While the method is efficient for certain datasets, it remains computationally demanding for larger datasets like the full ImageNet. Do the authors have plans to further reduce computational requirements in future work?

[1]. Wong E. et.al “Fast is better than free: Revisiting adversarial training” ICLR, 2020
[2]. Zhang Y. et.al “Revisiting and advancing fast adversarial training through the lens of bi-level optimization” PMLR, 2022
[3]. Wang Z. et.al “Preventing Catastrophic Overfitting in Fast Adversarial Training: A Bi-level Optimization Perspective” ECCV, 2024

---

> ### Author Response · Authors · 2024-11-22
>
> Thank you for your thoughtful comments and suggestions, and we try to address your questions and concerns below.
>
> **W1: The authors approach AT through the lens of continual learning but do not sufficiently establish the connection between the two ... despite the proposed method's effectiveness.**
>
> **A:** Thanks for your valuable feedback. We acknowledge the need to more concretely establish the connection between adversarial training (AT) and continual learning in our study. In response to your concerns, we have conducted additional experiments. To visually demonstrate the dynamic nature of adversarial sample distributions over the course of training, we employed t-SNE analysis [A]—a technique that effectively captures high-dimensional data complexity in a reduced dimensionality space. Specifically, we trained a PGD-10 model on the MNIST dataset using the PreActResNet-18 architecture under the  $\ell_{\infty}$ norm with $\epsilon=8/255$. We then performed t-SNE visualizations on the test adversarial samples at 20-epoch intervals, from the beginning to the end of training.
>
> The results, as shown in Figure 1 in ([link to Figure 1](https://anonymous.4open.science/r/xxx-xxx/iclr.pdf)) (If garbled, please refresh), illustrate distinct shifts in the data distribution of adversarial samples across epochs. This progression is similar with the challenges faced in continual learning, where models must adapt to evolving data distributions without forgetting previous states. This analysis visually substantiates the similarities between AT and continual learning processes.
>
> [A] Van et al. "Visualizing data using t-SNE."  JMLR  2008.
>
> **W2:  The AMS method requires tracking knowledge across multiple stages, which increases memory and computational costs, particularly with large datasets and models.**
>
> **A:** Thanks for your comment. AMS enhances model robustness by preventing forgetting, and we also agree that it introduces some overhead. However, the benefits in robustness could be significant for some scenarios, particularly for the security-critical applications. Also, our AMS approach is designed as a plugin that can be flexibly used to enhance existing adversarial training methods. So if a faster/more memory-efficient adversarial training method is developed in future, our AMS can be also plugged into it and the overhead could be alleviated correspondingly. Also please refer to our response to **Q4**.
>
> **Q1: Is the classification result shown in Figure 1 based on the training set? ... suggest similarity to catastrophic forgetting in continual learning.**
>
> **A:** Thank you for your question. The classification results shown in Figure 1 are based on the testing set.
>
> **Q2: Catastrophic overfitting is a common issue in fast adversarial training, ...  in continual learning?**
>
> **A:** Thank you for raising this question. In our experiments (Section 5.3), we observed an interesting phenomenon where our method effectively alleviates robust overfitting [B]. However, we have not specifically explored whether the AMS method could also address catastrophic overfitting in fast adversarial training, as this is not the primary focus of our paper. Nevertheless, we recognize the potential similarities and connections between catastrophic overfitting and catastrophic forgetting. We plan to conduct in-depth research on this topic in future work and appreciate your suggestion.
>
> [B] Rice et al. "Overfitting in adversarially robust deep learning". PMLR 2020.

---

> ### Author Response · Authors · 2024-11-22
>
> **Q3: Did the author consider dividing the epochs into stages unevenly, with each stage containing a different number of training epochs?**
>
> **A:** This is a good question. We considered dividing the training epochs into uneven stages. Specifically, we randomly selected $|\mathcal{M}|$ (where $|\mathcal{M}|= 200/m$) teacher models from the training process,  resulting in stages with variable numbers of epochs. To evaluate the impact of different stage division methods, we conducted the experiments and the results are shown in Table 1 below. The findings suggest that stage division methods have minimal impact on both clean and robust accuracy. For example, on the CIFAR-100 dataset, uneven method slightly increased robust accuracy by 0.3% compared to our even approach, whereas on CIFAR-10, it decreased robust accuracy by 0.3%.
>
> Table 1: Test accuracy (%) of TRADES and TRADES+AMS under the $\ell_{\infty}$ norm with $\epsilon=8/255$ based on the PreActResNet-18 architecture. The teacher models selection uses the different stage division methods
>
> | Method              | CIFAR-10 (clean) | CIFAR-10 (AA) | CIFAR-100 (clean) | CIFAR-100 (AA) |
> | ------------------- | ---------------- | ------------- | ----------------- | -------------- |
> | TRADES              | 83.25            | 49.11         | 58.26             | 25.93          |
> | TRADES+AMS (even)   | 84.86            | 51.22         | 60.03             | 28.05          |
> | TRADES+AMS (uneven) | 84.78            | 51.07         | 60.15             | 28.16          |
>
> **Q4: While the method is ... requirements in future work?**
>
> **A:** Thanks for this suggestion. Traditional adversarial training methods, such as PGD and TRADES, are computationally expensive, limiting their scalability for large datasets like the full ImageNet. Specifically, these methods require multiple backward propagations (BPs) per epoch to generate adversarial variants of all training data, followed by training the model with these adversarial examples. To address this issue, we plan to explore the integration of our AMS method with fast adversarial training techniques that significantly reduce the number of BPs required per data point. These techniques, referenced in [C-E], offer a promising direction for enhancing computational efficiency. In preliminary investigations, as noted in our response to **W1** of reviewer `VMqL`, we have already successfully integrated our AMS method with "Free adversarial training" [C], which can be accessed via ([link to pseudocode](https://anonymous.4open.science/r/xxx-xxx/iclr.pdf)). This integration has demonstrated potential in reducing computational demands. In future work, we aim to further explore the compatibility of AMS with other fast adversarial training methods.
>
> [C] Adversarial Training for Free. NeurIPS 2019.
>
> [D] You Only Propagate Once: Accelerating Adversarial Training via Maximal Principle. NeurIPS 2019.
>
> [E] Fast is better than free: Revisiting adversarial training. ICLR 2020.

---

> ### Author Response · Authors · 2024-11-27
>
> Dear Reviewer THQc,
>
> Thank you for taking the time to review our paper and provide valuable feedback. As the discussion phase is nearing its conclusion, we would like to confirm if our responses from a few days ago have effectively addressed your concerns. If you have any additional comments, we will do our best to address them.
>
> Best regards,
>
> The authors

---

> > ### Comment · Reviewer_THQc · 2024-12-02
> >
> > Thanks for the author's reply and additional experiments. Using continuous learning knowledge to solve the overfitting problem in adversarial training looks promising. I will improve my score.

---

> ### Author Response · Authors · 2024-12-02
>
> We thank the reviewer again for the positive and encouraging feedback.

---

### Official Review · Reviewer_Ezci · 2024-11-04

**Soundness:** 3
**Presentation:** 3
**Contribution:** 3
**Rating:** 6
**Confidence:** 4

**Summary:**

The paper studied the connection between forgetting (studied in the continual learning domain) and lack of adversarial robustness. The idea is that generated adversarial examples at every epoch has its own distribution (akin to tasks in continual learning literature), and learning on the new distribution (task) in the next epoch causes forgetting. The paper has demonstrated the forgetting phenomenon in adversarial training through experiments, and based on these observations, proposed a method called Adaptive Multi-teacher Self-distillation (AMS), which utilizes self-knowledge distillation from multiple checkpoints of the model throughout training to overcome forgetting.

**Strengths:**

1. The idea of studying forgetting in adversarial training is interesting and might be of interest to researchers in the field.
2. The paper is well-written, and it is easy to follow the text along with the experimental results.
3. Experimental results cover standard benchmarks in the field.
4. It is indeed interesting that the proposed method has been shown to be effective in mitigating robustness overfitting, and studying this direction could deepen researchers' understanding in the field.

**Weaknesses:**

1. The proposed loss function has multiple components, and each of them needs to be justified with ablation studies. In Section 4, where the loss function and algorithms are discussed, it is argued that "Reweighting-based Loss Correction" is needed; however, this is not demonstrated in the paper with experimental results.
2. From a memory perspective, multiple checkpoints of the model need to be stored and used for knowledge distillation, which is costly. This could be acceptable if the performance gain is notable; however, previous studies have shown higher performance gains by using only extra synthetic data (see point 5).
3. In the "Experimental Setting," it is stated that accuracy is reported for the best checkpoint of the model with the highest "test" accuracy, which is quite problematic.
4. The accuracy of the proposed method in Table 2 has been bolded, but for CIFAR-10, three of the baselines have higher accuracy than the proposed method; similarly for CIFAR-100.
5. Related to the previous point, other recent benchmarks, such as "Better Diffusion Models Further Improve Adversarial Training" (https://arxiv.org/abs/2302.04638), are missing in the table. For example, with 1M extra adversarial examples, this baseline achieves much higher robust accuracy than the proposed method with the same number of extra examples.
6. The study misses a comparable forgetting analysis in natural training. Although experiments have shown that forgetting occurs in adversarial training, it may not be specific to AT, and performing knowledge distillation only for the clean loss in TRADES might be sufficient. An experimental analysis to distinguish this could provide further insights.

**Questions:**

1. The main question is, if forgetting is the problem in adversarial training (AT), why does having more data (such as synthetic data) boost robust accuracy? If the number of samples increases, there are greater differences in the distributions of samples (tasks) from one epoch to another, which should potentially increase forgetting. However, in reality, robust accuracy improves. The explanation for this phenomenon appears to be missing from the paper, especially given that almost all recent SOTA methods utilize generated data, and even in the experimental section of this paper, a version with 1M extra generated samples is employed.

2. Regarding the lack of ablation study on the loss formulation, the hyperparameter sensitivity experiments indicate that the size of the interval parameter "m" has minimal effect on robustness performance, especially with larger values. If distribution shifts from one epoch to another cause forgetting, we would expect a decline in performance with larger values of "m," as less forgetting would be recovered; however, this trend does not appear in Figure 4. Why is this the case?


I have listed my concerns and questions in the weaknesses and questions sections and would appreciate it if the authors could provide precise justifications for them. However, in its current format, I am leaning toward rejection.

---

> ### Author Response · Authors · 2024-11-22
>
> Thank you for your thoughtful comments and suggestions, and we try to address your questions and concerns below.
>
> **W1: The proposed loss function ... that "Reweighting-based Loss Correction" is needed , ... with experimental results.**
>
> **A:**  Thank you for highlighting the necessity of an ablation study to validate the reweighting-based loss correction (RLC) in our methodology. We agree that such an analysis is important for demonstrating the specific contribution and impact of each component. In response to your suggestion, we have conducted the ablation experiments and included the results in the Table 1 below. These experiments specifically assess the impact of the RLC component on the robust and clean accuracies of models trained with TRADES+AMS and PGD-10+AMS. The results demonstrate that the inclusion of the RLC can enhance the robust accuracy across both CIFAR-10 and CIFAR-100 datasets to certain extent. For example, when RLC is removed, the robust accuracy on CIFAR-100 decreases from 28.05% to 27.62% compared with TRADES+AMS, and from 28.31% to 27.58% compared with PGD-10+AMS, illustrating the substantial contribution of the RLC component to enhancing adversarial robustness. Given the time limit of rebuttal, we plan to add more complete ablation study on RLC later.
>
> Table 1: Test accuracy (%) of TRADES+AMS and PGD-10+AMS with or without (w/o) the ''Reweighting-based Loss Correction'' (RLC) term under the $\ell_{\infty}$ norm with $\epsilon=8/255$ based on the PreActResNet-18 architecture.
>
> | Method             | CIFAR-10 (clean) | CIFAR-10 (AA) | CIFAR-100 (clean) | CIFAR-100 (AA) |
> | ------------------ | ---------------- | ------------- | ----------------- | -------------- |
> | TRADES+AMS         | 84.86            | 51.22         | 60.03             | 28.05          |
> | TRADES+AMS w/o RLC | 84.72            | 50.08         | 60.11             | 27.62          |
> | PGD-10+AMS         | 85.45            | 51.63         | 58.87             | 28.31          |
> | PGD-10+AMS w/o RLC | 85.66            | 50.31         | 58.83             | 27.58          |
>
> **W2: From a memory perspective ... need to be stored and used for knowledge distillation, which is costly ... using only extra synthetic data (see point 5).**
>
> **A:** Thank you for your comment about the memory costs associated with storing multiple checkpoints for knowledge distillation. We acknowledge that this approach incurs certain memory overhead compared to classic methods (e.g., PGD and TRADES).  As pointed out in references [1,2], using extra synthetic data can achieve substantial performance improvements primarily by enriching the training dataset. However, these methods primarily focus on enhancing robustness from the perspective of **data diversity** and do not address **model forgetting**. That is, our method and this synthetic data based method are mutually orthogonal, and they can be combined to achieve even better performance. We have conducted a comparative analysis of our method combined with those using extra synthetic data. The results, presented in Table 2 of our paper (lines 455 to 463), show that the robust accuracy can be improved by combining them together.
>
> [1]. Wang et.al “Better diffusion models further improve adversarial training.” ICML, 2023
>
> [2]. Gowal et.al “Improving robustness using generated data.” NeurIPS, 2021
>
> **W3: In the "Experimental Setting," it is stated that accuracyis reported for the best checkpoint of the model with the highest "test" accuracy, which is quite problematic.**
>
> **A:** Thanks for raising this point. We should revise it to "We report the test accuracy on the best checkpoint that achieves the highest robust **validation** accuracy under PGD-20".

---

> ### Author Response · Authors · 2024-11-22
>
> **W4: The accuracy of the proposed method in Table 2 has been bolded, but for CIFAR-10, three of the baselines have higher accuracy than the proposed method; similarly for CIFAR-100.**
>
> **A:** Yes, we acknowledge that several baseline methods show higher accuracy than our proposed method on CIFAR-10 and CIFAR-100. This discrepancy primarily stems from **different in Neural Network Architectures**. The baselines that outperform our method utilize different neural network architectures. For instance, the $S^2O$ method employs a more complex WideResNet-34-20 architecture, which inherently provides a better capacity for learning and generalization compared to the WideResNet-34-10 architectures used by our TRADES+AMS method.
>
> Our AMS approach is designed as a plugin that enhances existing adversarial training methods. In our study, we demonstrate this by combining AMS with TRADES. However, AMS's flexibility allows it to be combined with other robust training methods, including $S^2O$. To address your concern, we conducted additional experiments where AMS was combined with $S^2O$. The results demonstrate that combining AMS with $S^2O$ can enhance the robust accuracy and meanwhile preserve high clean accuracy. Specifically, on the CIFAR-100 dataset, the robust accuracy is improved from 27.50% to 29.96%.
>
> Table 2: Test accuracy (%) of $S^2O$ and $S^2O$ + AMS under the $\ell_{\infty}$ norm with $\epsilon=8/255$ based on the **WideResNet-34-10**  architecture.
>
> | Method       | CIFAR-10 (clean) | CIFAR-10 (AA) | CIFAR-100 (clean) | CIFAR-100 (AA) |
> | ------------ | ---------------- | ------------- | ----------------- | -------------- |
> | $S^2O$       | 85.58            | 53.93         | 63.31             | 27.50          |
> | $S^2O$ + AMS | 85.83            | 55.92         | 63.67             | 29.96          |
>
> **W5: Related to the previous point, ..."Better Diffusion Models Further Improve Adversarial Training"  are missing in the table. ... of extra examples.**
>
> **A:** Thanks for mentioning this method. Actually we compared our method with this method (Wang et al., 2023; 1M) in Table 2 of our manuscript (lines 455 to 463).
>
> **W6: The study misses a comparable forgetting analysis in natural training ... provide further insights.**
>
> **A:** We appreciate your suggestion for distinguishing the effects of forgetting in adversarial training (AT) from natural training scenarios.
>
> **Reason for Omitting Natural Forgetting Analysis:** In Section 3 of our paper, we analyze that the fundamental cause of forgetting in AT is attributed to the significant shift in the distribution of adversarial samples, a phenomenon not typically observed with natural samples throughout the training process. Given this, we focused our analysis on adversarial settings where the impact of forgetting is more pronounced.
>
> **Experiment on Knowledge Distillation for Clean Loss in TRADES:** To address your concern, we conducted additional experiments. The results are summarized in Table 3 below. The results indicate that while enhancing the clean loss through knowledge distillation (TRADES+KDCL) does lead to some improvements in both clean and robust accuracies, the gains are not that significant, compared to our proposed method (TRADES+AMS). Specifically, TRADES+AMS consistently shows superior performance in robust accuracy across both datasets. This suggests that our approach is more effective in adversarial training.
>
> Table 3: Test accuracy (%) of TRADES, TRADES+AMS and TRADES with "knowledge distillation only for the clean loss" (KDCL) under the $\ell_{\infty}$ norm with $\epsilon=8/255$ based on the PreActResNet-18 architecture.
>
> | Method      | CIFAR-10 (clean) | CIFAR-10 (AA) | CIFAR-100 (clean) | CIFAR-100 (AA) |
> | ----------- | ---------------- | ------------- | ----------------- | -------------- |
> | TRADES      | 83.25            | 49.11         | 58.26             | 25.93          |
> | TRADES+KDCL | 84.93            | 49.84         | 60.05             | 26.71          |
> | TRADES+AMS  | 84.86            | 51.22         | 60.03             | 28.05          |

---

> ### Author Response · Authors · 2024-11-22
>
> **Q1: The main question is, ...why does having more data (such as synthetic data) boost robust accuracy? ... a version with 1M extra generated samples is employed.**
>
> **A:** Thank you for your insightful question. We appreciate the opportunity to clarify this important point.  The primary reason that adding more data, including synthetic data, can boost robust accuracy is that it enriches the training set with a wider variety of examples. This diversity helps the model learn more generalized and robust features, making it more capable to handle adversarial attacks. But this improvement does not mean that there is no forgetting during the adversarial training. Our comparative experiments (Table 2 in page 9) also confirm this point. For example, combining our AMS with Wang et al. (2023) (1M) on the SVHN dataset improves the robust accuracy from 61.73% to 63.15%.
>
> So we think although adding synthetic data can boost robust accuracy, the forgetting issue still exists. But the benefit gained from adding synthetic data may outweigh the impact from forgetting. If we can alleviate the forgetting during the “synthetic data involved” adversarial training, the improvement could be even more significant, just like the experimental result shown in Table 2 of page 9. Overall, we think adding more training data to boost performance does not conflict with the existence of forgetting.
>
>
>
> **Q2: Regarding  ... that the size of the interval parameter "m" has minimal effect on robustness performance, especially with larger values... in Figure 4. Why is this the case?**
>
> **A:** Thanks for raising this question. We believe the reason why the trend does not appear in Figure 4 is due to the diminishing significance of **cumulative effects** as $m$ increases. Specifically, the impact of interval frequency on the total number of checkpoints becomes less significant with larger intervals. For example, within a total of 200 training epochs: Saving a checkpoint every 40 epochs results in approximately 5 checkpoints. Saving every 50 epochs results in 4 checkpoints. The difference in the number of checkpoints between these two settings is minimal, hence the marginal impact on robustness. However, for smaller values of "m", such as every 10 or 20 epochs, the number of checkpoints increases significantly (to 20 and 10 checkpoints, respectively), introducing more opportunities to alleviate forgetting through these frequent snapshots. As $m$ increases, the robustness performance tends to stabilize rather than continuously decline.

---

> ### Author Response · Authors · 2024-11-27
>
> Dear Reviewer Ezci,
>
> Thank you for taking the time to review our paper and provide valuable feedback. As the discussion phase is nearing its conclusion, we would like to confirm if our responses from a few days ago have effectively addressed your concerns. If you have any additional comments, we will do our best to address them.
>
> Best regards,
>
> The authors

---

> > ### Comment · Reviewer_Ezci · 2024-11-27
> >
> > Dear Authors,
> >
> > Thank you for providing a detailed response. I believe most of my concerns have been addressed, and I am happy to increase my score.

---

> ### Author Response · Authors · 2024-11-28
>
> We thank the reviewer again for the helpful comments.

---

### Author Response · Authors · 2024-11-24
**General Response**

We sincerely thank all the anonymous reviewers for their valuable and insightful comments. We have added some content to our revised paper according to the comments and questions. The changes are summarized below and marked in ***red*** in the paper.

#### Revisions to the paper

- *page 2, Line 82 -89*; We have added a reference [1] and a discussion to clarify the relevance of our paper to it (for **W6** of reviewer `XV9q`).
- *page 5 and 20, Appendix B.2;* We added a new Appendix B.2 to show the connection between AT and CL (for **W1** of reviewer `THQc`).
- *page 8, Line 379, 395, 427*; We added some necessary references and explanations, and also corrected "test" to "validation" (for **W3** of reviewer `Ezci`).
- *page 9, Table 2;* We added some supplementary experiments and adjusted the format in Table 2. In order to unify the comparison on WideResNet, we added an experiment of AWP[2]+AMS on WideResNet-34-10 (Line 451) (for **W1** and **W2** of reviewer `XV9q`; for **W2** of reviewer `rCAv`).
- *page 10 and 25, Section 5.5;* We expanded our experimental setup and results to include ViT and VGG16 architectures (for **W2** of reviewer `VMqL`).
- *page 10 and 25, Section 5.6 and Appendix B.7;* We added a new Section and Appendix to conduct an ablation experiment (for **W1** of reviewer `Ezci`; for **W3** of reviewer `rCAv`).
- *page 19, Line 974-980;* We added a summary of each subsection.
- *Page 9, Line 481:* We moved a table from Section 5.4 (Table 4 in the original version) to Appendix B.5 (Table 8 in the revision).

[1] Improving the affordability of robustness training for DNNs, CVPR 2020

[2] Adversarial Weight Perturbation Helps Robust Generalization, NeurIPS 2020

---

### Meta-Review · Area_Chair_vuTR · 2024-12-23

**Metareview:**

This paper studies an interesting connection between continual learning and adversarial robustness. The idea is that generated adversarial examples at every epoch has its own distribution, and learning on the new distribution in the next epoch causes forgetting. Based on this observation, the authors propose a novel method that distills the knowledge from multiple teachers (different checkpoints) during adversarial training. Experiments on multiple datasets show solid performance improvement.

**Additional Comments On Reviewer Discussion:**

During the rebuttal, the reviewers raised several concerns including missing ablation studies, missing experiments on ViT architectures, memory and compute overhead, etc. The authors provided good responses to most questions, adding several important experiments. Overall, I think the paper is in a good shape now. Post rebuttal, all reviewers lean towards accepting the paper. I think this is a good paper that deserves to be accepted.

---

### Decision · Program_Chairs · 2025-01-22

Accept (Poster)